# GADF-VGG16 based fault diagnosis method for HVDC transmission lines

Hao Wu[1,2]ʘ*, Yuping Yang[1,2]ʘ, Sijing Deng[1,2]‡, Qiaomei Wang[3]‡, Hong Song[4]

**1** Artificial Intelligence Key Laboratory of Sichuan Province, Zigong, Sichuan Province, China, **2** Automation and Information School of Automation and Information Engineering, Sichuan University of Light Chemical Technology, Zigong, Sichuan Province, China, **3** Chengdu Institute of Technology, Chengdu, Sichuan Province, China, **4** Aba Teachers College, Wenchuan, Sichuan Province, China

ʘ These authors contributed equally to this work.
‡ SD and QW also contributed equally to this work.
* wuhao801212@163.com

## Abstract

Transmission lines are most prone to faults in the transmission system, so high-precision fault diagnosis is very important for quick troubleshooting. There are some problems in current intelligent fault diagnosis research methods, such as difficulty in extracting fault features accurately, low fault recognition accuracy and poor fault tolerance. In order to solve these problems, this paper proposes an intelligent fault diagnosis method for high voltage direct current transmission lines (HVDC) based on Gramian angular difference field (GADF) domain and improved convolutional neural network (VGG16). This method first performs variational modal decomposition (VMD) on the original fault voltage signal, and then uses the correlation coefficient method to select the appropriate intrinsic mode function (IMF) component, and converts it into a two-dimensional image using the Gramian Angular Difference Field(GADF). Finally, the improved VGG16 network is used to extract and classify fault features adaptively to realize fault diagnosis. In order to improve the performance of the VGG16 fault diagnosis model, batch normalization, dense connection and global average pooling techniques are introduced. The comparative experimental results show that the model proposed in this paper can further identify fault features and has a high fault diagnosis accuracy. In addition, the method is not affected by fault type, transitional resistance and fault distance, has good anti-interference ability, strong fault tolerance, and has great potential in practical applications.

## 1 Introduction

China's energy base is far from the load-intensive center, so trans-regional power transmission is a necessary means to realize the reasonable allocation of resources [1]. High voltage direct current transmission (HVDC) is widely used by virtue of its large transmission capacity, low line loss, and long transmission distance [2]. Due to the long distance spanned by the HVDC line, the unfavorable geographical locations and the harsh climate environment, the probability of line failure is high. According to statistics, about 50% of the faults in the DC transmission

**Data Availability Statement:** All relevant data are within the paper and its Supporting Information files.

**Funding:** This research was supported by the following founders: (1)The Project of Sichuan

provincial science and Technology Department (Grant No. 2018JY0386, 2020YFG0178, 2021YFG0313); (2)The artificial intelligence key laboratory of Sichuan province Foundation (2019RYY01); (3)Enterprise informatization and Internet of things measurement and control technology key laboratory project of Sichuan provincial university (2018WZY01, 2019WZY02, 2020WZY02); (4)The Project of Sichuan Provincial Academician (Expert) workstation of Sichuan University of Science and Engineering (2018YSGZZ04); (5)The Project of Zigong Science and Technology Bureau (2019YYJC13, 2019YYJC02, 2020YGJC16). The funders had no role in study design, data collection and analysis, decision to publish, or preparation of the manuscript.

**Competing interests:** The authors have declared that no competing interests exist.

system are DC line faults, so research on the correct detection and diagnosis of the fault is of great significance for the safe and reliable operation of the power system [3].

At present, experts and scholars have proposed a series of protection schemes for the protection of HVDC transmission lines. Among them, traveling wave protection and differential under voltage protection are generally used as the main protection methods, and longitudinal differential protection and low voltage protection as backup ones [4]. Among them, the traveling wave protection principle is most widely used in many protection methods.

For example, in reference [5], based on the double-ended traveling wave method, it is proposed to use the time interval between the detection of the first incident traveling wave and the reflected wave at the detection point for the identification of faults occurred within and outside the protection zone. However, the method relies on the transmission of double-ended information, which has an impact on the quickness of the protection. Considering that traveling wave protection is susceptible to high transitional resistance and fault distance, the author of reference [6] proposes to use traveling wave transmission principle combined with Teager energy operator to differentiate the internal fault from the external fault in the HVDC system. In this method, fault identification can be completed in a short time and the fault type can also be accurately identified under high resistance faults at the remote end. However, how to accurately identify the traveling wave head in the traveling wave method is an insurmountable technical problem [7]. In reference [8], a distributed parameter model-based fault identification scheme for HVDC transmission lines inside and outside the protection zone is proposed. This method does not rely on the traveling wave protection principle, and the accuracy of its fault detection depends on the setting of parameters of the transmission line.

In recent years, a large number of scholars have proposed to use support vector machines (SVM), Back propagation neural networks (BP), artificial neural networks (ANN), random forests and other methods [9–12] to study the problem of transmission line fault diagnosis. For example, Johnson et al [9] used the SVM classification mechanism to achieve fault identification and classification of HVDC transmission lines. The feature vector used in the classification modules is the standard deviation of the signals over half cycle before and after the occurrence of fault. However, the method does not fully exploit the waveform features of the faulty traveling waves and the fault tolerance of the method needs to be further verified.

R. Kou et al [10] used the amount of electrical variation in transmission line fault conditions as a feature vector to train BP neural networks for fault diagnosis. This method has a good identification effect for faults occurred within the protection zone, but fails to take the identification of faults occurred outside of the protection zone into account. K. Moloi et al [11] used a particle swarm algorithm (POS) to optimize an artificial neural network (ANN), and then used the optimized ANN model to identify and classify different faults.This method can identify faults occurred within and outside of the protection zone with an accuracy of 99%. However, whether the method can also correctly identify fault types in high transitional resistance and high noise environments remains to be proven. Wu et al [12] used random forest (RF) neural networks to achieve the selection of fault poles and identification of fault types on HVDC transmission lines. But, the extraction of fault features is more complex with this method.

As a research hotspot in recent years, deep learning has been widely used in image processing, target detection, fault diagnosis and other fields. For example, Jin et al. [13] introduced an intelligent fault diagnosis method for train axle box bearings based on parameter optimization of VMD and improved DBN. In this method, the gray wolf optimization algorithm (GWO) is used to optimize VMD and DBN, and the problem of VMD parameter setting is solved. At the same time, VMD is used to decompose the signal, and the feature information of the modal component with the largest correlation coefficient is extracted through multi-scale distribution

entropy. Finally, DBN is used to identify the feature information to realize the weak fault diagnosis of bearings. Reference [14] proposed an intelligent diagnosis algorithm based on continuous wavelet transform and Gaussian convolution deep belief network. In this method, the one-dimensional vibration signal is converted into a two-dimensional image and the deep belief network is used for feature extraction, which avoids the influences of artificial extraction of fault features. Smart algorithms have achieved significant improvements over traditional algorithms. Reference [15] proposed a new method for intelligent fault diagnosis based on time-frequency images and deep learning. This method uses short-time Fourier transformation to obtain time-frequency images, and combines the multi-sensory data fusion method of deep residual network to carry out the analysis of machine bearings. For fault diagnosis, compared with a single type of fault signal, this method can achieve better diagnostic accuracy. The development of deep learning has also brought new ideas to the field of fault diagnosis of transmission lines. Reference [16] proposes a fault diagnosis method for HVDC systems based on stacked sparse autoencoders, which utilizes stacked sparse autoencoders for automatic feature extraction, classification and identification. This method makes full use of the self-learning ability of the deep learning algorithm to realize the accurate identification of regional faults. Zhai et al [17] proposed to construct a fault diagnosis model for HVDC transmission lines using an improved convolutional neural network (CNN) to extract features and implement fault classification for current timing data. Compared with the traditional CNN network, the method has a certain improvement in recognition accuracy, but the CNN is not ideal for feature extraction of time-series signals. Therefore, drawing on the research idea of converting time series signals into two-dimensional images and combining deep learning algorithms in the field of bearing fault diagnosis, many scholars have begun to try to use this research method in fault diagnosis of HVDC transmission lines. For example, Wang Jun et al. [18] proposed to convert one-dimensional time series signals into two-dimensional grayscale images, and then use CNN to classify transmission line faults. However, the process of converting time series signal into grayscale image in this method results in the data loss of feature. In summary, it is feasible to apply deep learning to fault diagnosis of transmission lines, and further in-depth research is needed.

In view of the advantages and disadvantages of the above fault diagnosis algorithms, this paper adopts the deep learning method to propose a fault diagnosis model for HVDC transmission lines based on GADF-VGG16. The fault voltage signal of the HVDC transmission line is decomposed into modal components by VMD, and the selected IMF modal components are converted into color images through the Gramian Angular Difference Field (GADF), and the images are input into the improved VGG16 for feature extraction and classification.This method uses a novel GADF encoding method for data preprocessing, and constructs a bijective map for one-dimensional time series and two-dimensional space series, which will not cause the loss of feature information [19].At the same time, this paper improves the traditional VGG16 model structure, increases the BN layer and the convolution layer of the dense connection structure, speeds up the training and convergence speed of the network, and realizes the reuse and enhancement of features. Meanwhile, the global average pooling layer is used to replace the fully connected layer, which reduces the amount of model parameters and computing time, making the proposed method more suitable for rapid fault diagnosis. The experimental results show that the fault diagnosis model proposed in this paper has high accuracy, is not susceptible to fault type, transitional resistance and fault distance, and has good anti-interference ability and fault tolerance.

The main contributions of this paper are as follows:

1. Based on the GADF algorithm, the one-dimensional fault signal is converted into a two-dimensional color image, aiming to use deep learning to further explore fault feature information.

2. The batch normalization algorithm is introduced to effectively speed up network convergence and prevent overfitting. The dense connection method of the convolutional layer is designed to speed up the network training and convergence speed while realizing feature reuse and enhancement. Aiming at the slowing down of the calculation speed caused by the large amount of network parameters, a global average pooling algorithm is introduced, which effectively reduces the amount of network parameters.

3. The fault diagnosis model of GADF-VGG16 is proposed, which improve the fault diagnosis accuracy, and the superiority of the proposed algorithm is verified by an example.

## 2 Two-dimensional image expression based on fault signal

### 2.1 Simulation model of HVDC transmission system

Referring to a domestic HVDC transmission project, this paper uses the PSCAD-EMTDC simulation platform to build a model of HVDC transmission system. The model parameters are as follows: transmission power is 3000MW, rated voltage is 500kV, rated current is 3kA, DC filter is 12/24/36 three-tuned filter, tower structure adopts DC2 tower commonly used in engineering, and the set line length is 1000km. The simulation model is shown in Fig 1.

The boundary element is composed of a smoothing reactor and a DC filter, and the protection equipment is installed inside the boundary element on the rectifier side. The fault simulation of the model is carried out. Take point F1 as an example for external single-pole-to-ground fault (abbreviated as EG), point F2 as an example for positive single-pole-to-ground fault in the zone (abbreviated as PG), point F3 as an example for negative single-pole-to-ground fault in the zone (abbreviated as NG), and point F4 as an example for double-line-to-ground fault in the zone (abbreviated as PNG), and collects the fault voltage signals under four fault conditions. The PSCAD is running to generate data under above four fault conditions.

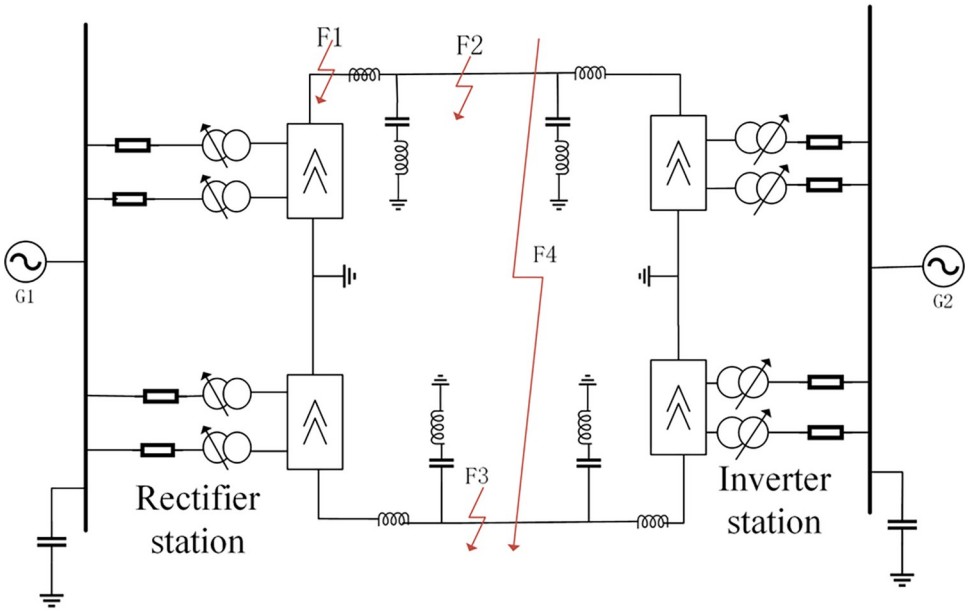

**Fig 1. HVDC Transmission system model.**

## 2.2 Preprocessing the original data

In this paper, the VMD decomposition algorithm is used to preprocess the original fault signal, and K IMF components with center frequencies are obtained. Then we refer to the method mentioned in reference [20] to estimate the value of K using center frequency rules. In this paper, the number of modal decompositions is set to be K = 6.

The correlation coefficient is a statistical analysis index that reflects the degree of rank correlation, and its formula is defined as

$$P_{XY} = \frac{Cov(X, Y)}{\sqrt{D(X)}\sqrt{D(Y)}} = \frac{E(X - EX)(Y - EY)}{\sqrt{D(X)}\sqrt{D(Y)}} \tag{1}$$

From Formula (1), $P_{XY}$ is the correlation coefficient representing the variable X and the variable Y, X and Y represent two independent variables, E represents the mean, D represents the variance, and Cov represents the covariance. The value range of $P_{XY}$ is [-1, 1]. The larger the value, the higher the correlation with Y, and the strong correlation between X and Y. Taking the fault at point F2 in the zone (fault distance 10km, transition resistance 150Ω) as an example, calculates the correlation coefficient between the original signal X and each IMF component $Y_i$ (i = 1,2. . . . . .6) decomposed by VMD. The calculation results are shown in Table 1.

Reference [21] proposed that the greater the correlation coefficient between the modal component and the original signal, the higher the similarity between the two, that is, the more fault information the modal component contains. When selecting the modal components, it should be noted that the modal components should not only contain the fault characteristic frequency, but also cannot be too close to the original signal, otherwise the significance of decomposition will be lost [22]. Analysis of Table 1 shows that if the correlation coefficient is greater than 0.6, which indicates that IMF2 is strongly correlated with the original signal. Therefore, this paper selects IMF2 component as the characteristic signal.

## 2.3 GADF image encoding

The Gram Corner Field (GAF) is a novel time series encoding method proposed by Zhiguang Wang and Time Oates in 2015 [23]. The specific implementation steps are as follows:

(1) Normalize the one-dimensional time series, and scale the value to between

[−1,1].Therefore we have:

$$\tilde{x}_i = \frac{(x_i - x_{min}) + (x_i - x_{max})}{x_{max} - x_{min}} \tag{2}$$

(2)Convert the normalized time series to polar coordinates, use the angle to represent the value of the time series, and the radius to represent the timestamp:

$$\begin{cases} \theta_i = \arccos(\tilde{x}_i) \\ r_i = \dfrac{t_i}{N} \end{cases} \quad -1 \le x_i \le 1, \tilde{x}_i \in \tilde{X} \tag{3}$$

It can be seen from Formula (3) that the value range of θ is between [0, π], and cosθ is

**Table 1. Correlation coefficient values between different IMF components and the original signal.**

| serial number | $P_{XY_1}$ | $P_{XY_2}$ | $P_{XY_3}$ | $P_{XY_4}$ | $P_{XY_5}$ | $P_{XY_6}$ |
|---|---|---|---|---|---|---|
| Correlation coefficient | 0.881 | 0.782 | 0.399 | 0.234 | 0.101 | 0.064 |

monotonic on [0, π]. Therefore, the mapping of a given time series in the polar coordinate system is unique [21]. While N represents the constant factor of the space generated by the regularized polar coordinate system, and ti represents the timestamp. The time dependence is thus maintained by the r coordinate.

(3) GAF coding obtains the Gram angle difference field (GADF) through the trigonometric function difference operation as follows:

$$GADF = [\sin(\theta_i - \theta_j)] = \sqrt{I - \tilde{X}^2}' \bullet \tilde{X} - \tilde{X}' \bullet \sqrt{I - \tilde{X}^2} \tag{4}$$

Where $I$ is a unit row vector, and $\tilde{X}$ and $\tilde{X}'$ are the row vector of the sequence before and after scaling,and $\theta_i\{i = 1,2,...,n\}$ is the angle between the two vectors. Because the values of the time series obey the uniform distribution [−1, 1], the encoded images have great sparsity. Therefore, converting a one-dimensional time series signal into a GADF image increases the sparsity of the data, eliminates multimodal redundant information, and weakens the nonlinearity of the data.

Taking the four fault situations of F1 to F4 in Fig 1 as an example, the collected original fault signal is decomposed by VMD, and IMF2 is selected as the waveform diagram of the fault characteristic signal, as shown in Fig 2(A), 2(C), 2(E), 2(G) shown. Then, the IMF2 components are converted into two-dimensional images by GADF coding [24], as shown in Fig 2(B), 2(D), 2(F), 2(H).

## 3 Improved VGG16 network structure

### 3.1 VGG16 model

The VGG16 network is a new convolution neural network structure proposed by Simonyan et al of Oxford University [25]. This network mainly improves the structure of the feature extraction part of the traditional convolutional neural network, and reduces the calculation amount of the neural network by using multiple small convolution kernels instead of large convolution kernels, and does not reduce the receptive field of the network. However, the VGG16 network full connection layer parameters are too large, which leads to a large memory consumption, and also leads to a large amount of network calculation and a long training time. The network structure is shown in Fig 3. In view of the shortcomings of the network, this paper proposes an improved VGG16 model.

### 3.2 Improved VGG16 network

In order to improve the VGG16 network's ability to diagnose fault signals, a fault diagnosis model based on improved VGG16 is established in this paper [26]. First, a batch normalization layer is added to normalize the training data, which is conducive to speeding up the training speed of the network; Secondly, the dense connection structure is used to improve the connection method of the convolution block to realize feature reuse and better use of shallow feature information to avoid feature loss; finally, global average pooling is used to replace the fully connected layer in the network, which effectively reduces the amount of network parameters and improves network computing speed. Finally, the effect of improving the network's ability to identify the faults of DC transmission line correctly and rapidly is achieved.

**3.2.1 Add batch normalization(BN) layer.** When training a deep network, it is necessary to adjust the network parameters. The adjustment of parameters such as parameter initialization, weight coefficients, and learning rates plays a crucial role in the training of the entire network.

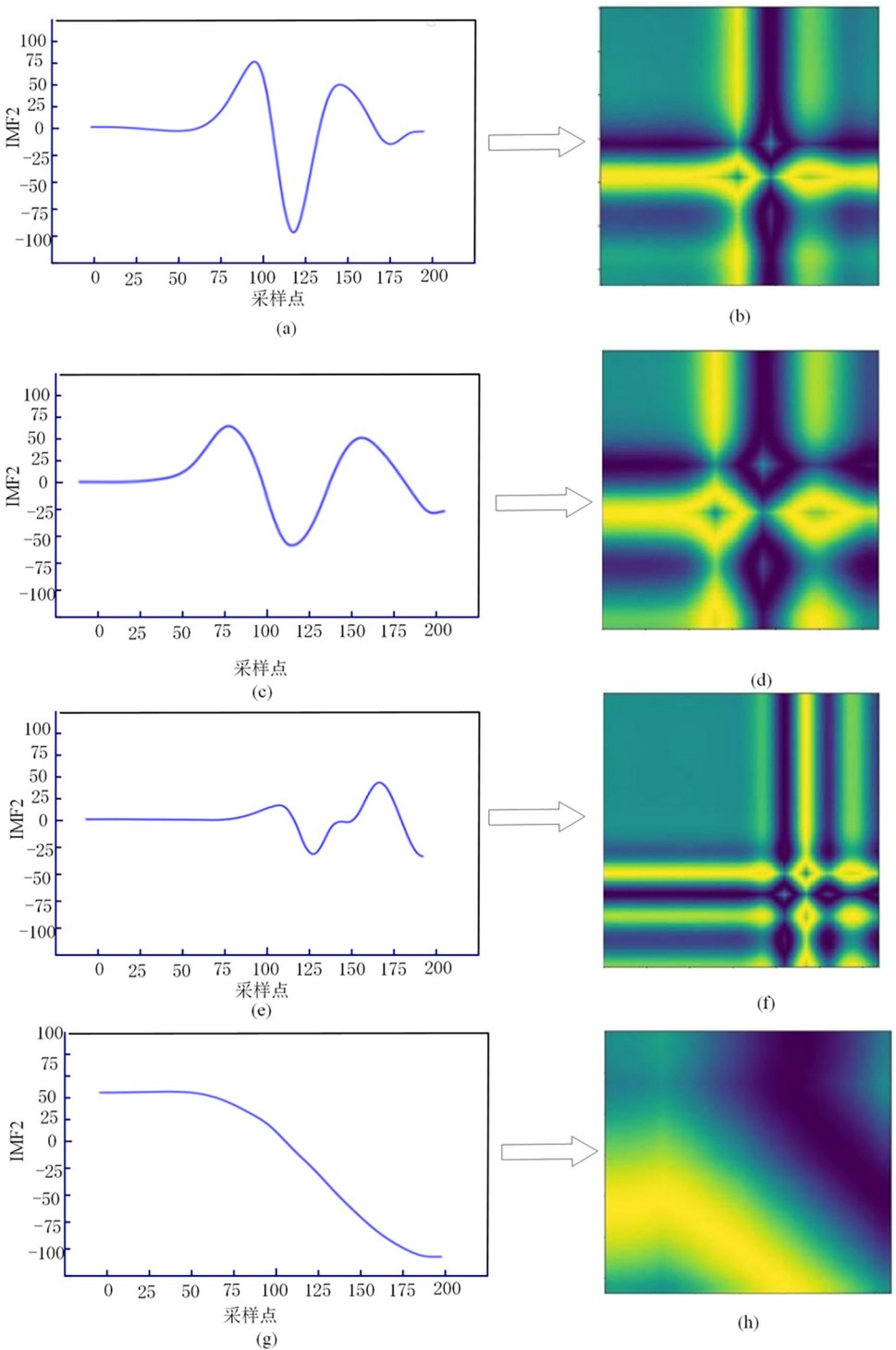

**Fig 2. Comparison of IMF component waveforms and GADF coding diagrams of different types of faults.**

In order to simplify the parameter adjustment process, Ioffe et al. [27] proposed the BN algorithm for the first time, and batch normalized the output data of each hidden layer in the training process, which reduces the influence of the hidden layer data distribution change on

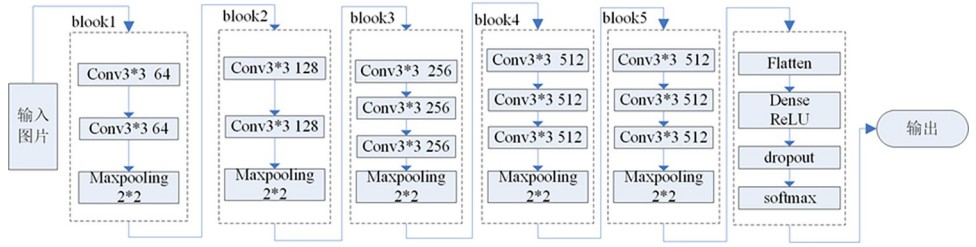

**Fig 3. Primeval VGG16 Structural model.**

the network, reducing the dependence of the neural network on parameters, speeding up the training speed of the neural network, and improving the stability of the network.

This paper introduces a batch normalization layer to improve the network model structure, as shown in Fig 4. A BN layer is added to each ordinary convolution module. Normalize the input to the activation function, which solves the effect of shifting and increasing the input data.

**3.2.2 Use dense connection.** For deep neural networks, if a single connection method is used between convolutional layers, with the deepening of the network, it is difficult for deep convolutions to obtain image features of shallow convolutions, which results in the loss of features between convolutional layers. In response to this problem, Gao [28] et al. proposed the DenseNet structure, which established dense connections between all the previous layers and the latter layers, and realized short-circuit connections through concatenate features, thereby realizing feature reuse and effectively solving the problem of feature loss. However, due to the dense connection of DenseNet, each layer will fuse the feature information of all previous layers, resulting in feature redundancy in the deep network, leading to high internal access cost and energy consumption [29]. Considering the problem of cost and internal friction, Lee [30] et al. proposed the VOVNet network, which adopts the one-shot aggregation (OSA) module to optimize the connection method of DenseNet, and only aggregates all the previous layers in the last layer at one time, thus effectively avoiding the disadvantage of feature redundancy, as well as making the feature utilization of each layer more efficient. This paper draws on the dense connection method of VOVNet, and adopts the OSA connection mode for the convolutional layers in the last three convolutional blocks (blook3~blook5) of the network. Taking blook3 in Fig 5 as an example, the convolutional layers c1, c2, and c3 are aggregated at the last layer through the concatenate operation to realize the reuse of features, and finally use 1×1 convolution layer to adjust the number of channels, as shown in Fig 5.

**3.2.3 Global average pooling(GAP) instead of fully connected layer.** Due to the large number of parameters in the fully connected layer of the convolutional neural network, the network is bloated and easy to cause over-fitting. Reference [30] first proposed the idea of replacing the fully connected layer with the global average pooling layer. GAP is to remove the characteristics of the black box in the fully connected layer, and directly give each channel the actual category meaning. The implementation process is shown in Fig 6(A).

This paper draws on the idea of reference [30], and uses GAP to replace the fully connected layer in the original model. In order to maintain the structural consistency with the original VGG16 model, three layers of 1×1 convolution layers are added. Finally, use the softmax layer to achieve classification. Its structure is shown in Fig 6(B). The purpose of using 1×1 convolutional layer is to increase the nonlinearity of the network while keeping the dimension of the feature map unchanged [31]. In addition, Fig 6(B) shows that the improved model similar to

# 3*3conv

# BN

# RELU

**Fig 4. Convolution module added to the BN layer.**

the original network in the structure. What's more, verified by experiments, by adding the 1×1 convolutional layers the performance of the network is better, and the final classification accuracy is higher.

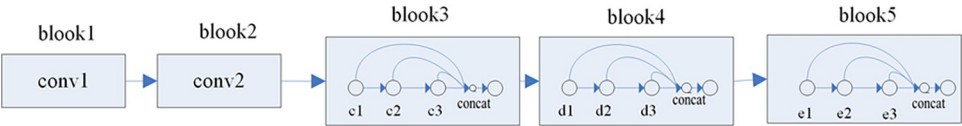

**Fig 5. Improved convolutional network structure.**

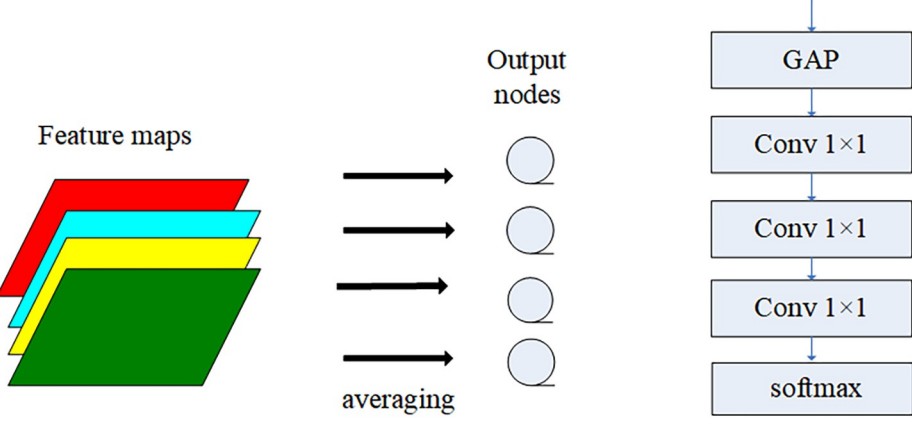

(a)GAP implementation process diagram    (b) Improved fully connected layer structure

**Fig 6. Global average pooling instead of fully connected layer.**

## 4 Fault diagnosis model of HVDC transmission line based on GADF-VGG16

The process of the GADF-VGG16 HVDC transmission line fault diagnosis method proposed in this paper is as follows, and the overall flow chart of the protection scheme is shown in Fig 7:

1) Train the improved VGG16 model and save the network parameters:

①First obtain the original data of fault voltage through PSCAD simulation;

②The original data is preprocessed by VMD decomposition, and IMF2 is selected as the feature signal for subsequent processing;

③Perform GADF encoding on IMF2 to convert one-dimensional signals into two-dimensional images;

④The obtained image samples are divided into training set and test set, and the training set is sent to the improved VGG16 proposed in this paper for training. The network training is completed and the network model is saved.

2) Test the improved VGG16 model:

①After step 1) get the test set

②Input the test set into the trained and improved VGG16 for testing

3) VGG16 performs fault identification and pole selection;

4) According to the judgment result, the corresponding protection action is realized, and the protection ends.

## 5 Experimental research and analysis

### 5.1 Experimental environment and data

This experiment was carried out on a deep learning workstation. The hardware configuration of the workstation is as follows: the CPU is Intel Xeon Silver 4210, the GPU is RTX 2080Ti,

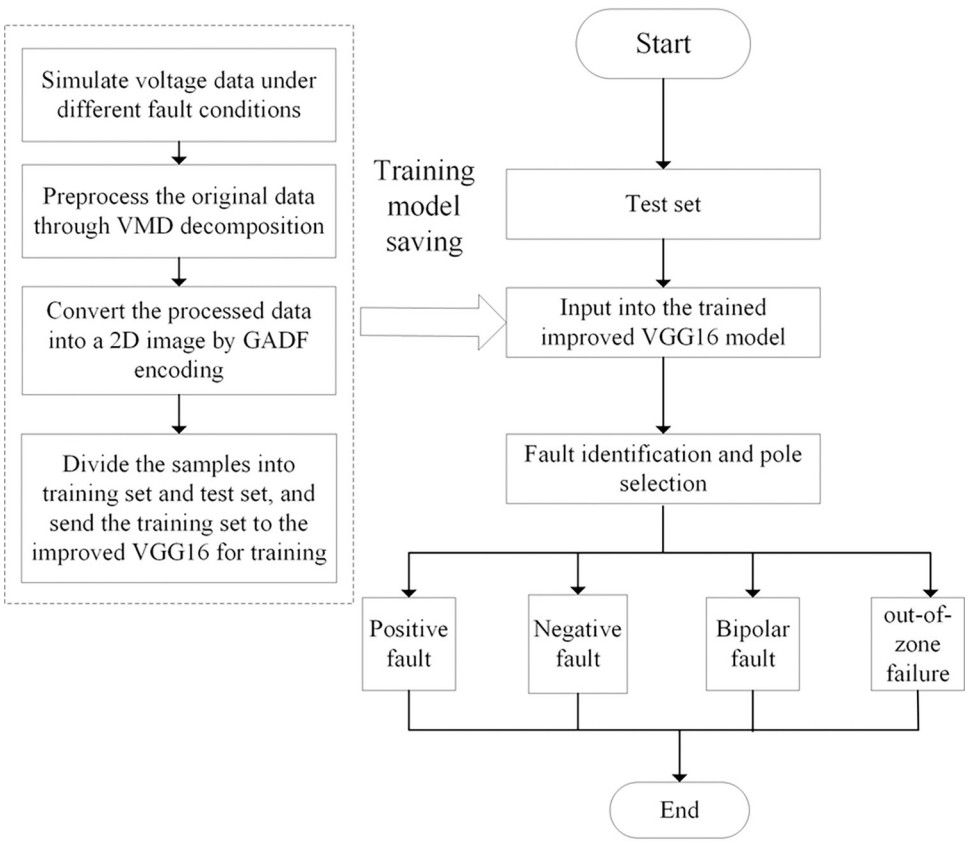

**Fig 7. Overall flow chart of the protection plan.**

and the memory is 64G; the software configuration is Anaconda, Pycharm; the programming language is Python, and the deep learning framework is Keras.

The sample data in this paper is composed of the failure data under four conditions as shown in Fig 1(internal positive fault (PG), internal negative fault (NG), internal bipolar fault (PNG), and external fault (EG).). The sampling frequency is 10kHz, and the time window is 10ms. Though simulation, we collect 1026 sets of sample datas, with the 210 groups of PG faults,210 groups of NG faults, 210 groups of PNG faults and 396 groups of EG faults. And then converting this sample data into 1026 pictures by GADF,of which 25 images were selected from each of the four types of faulty 2D images, making a total of 100 images to form the test set, leaving 926 images as the training set.

## 5.2 Network training

During network training, the input picture size is 128×128. The network is trained for 200 epochs, the batch size is set to 32, and the learning rate is set to 0.0001. Using the learning rate descent strategy, if the loss value for two consecutive rounds of validation sets is not improved, and the learning rate will be multiplied by 0.5.

Figs 8 and 9 are the Loss change curve and accuracy change curve obtained after 200 epoch of the improved VGG16 model training.From the analysis of Figs 8 and 9, it can be seen that the accuracy rate starts to stabilize at 100 epoch,the accuracy of the training set has reached 99.01%, and the accuracy of the testing set has reached 100%.After that, the accuracy and loss tended to be stable, and the model converged completely.The result shows that the

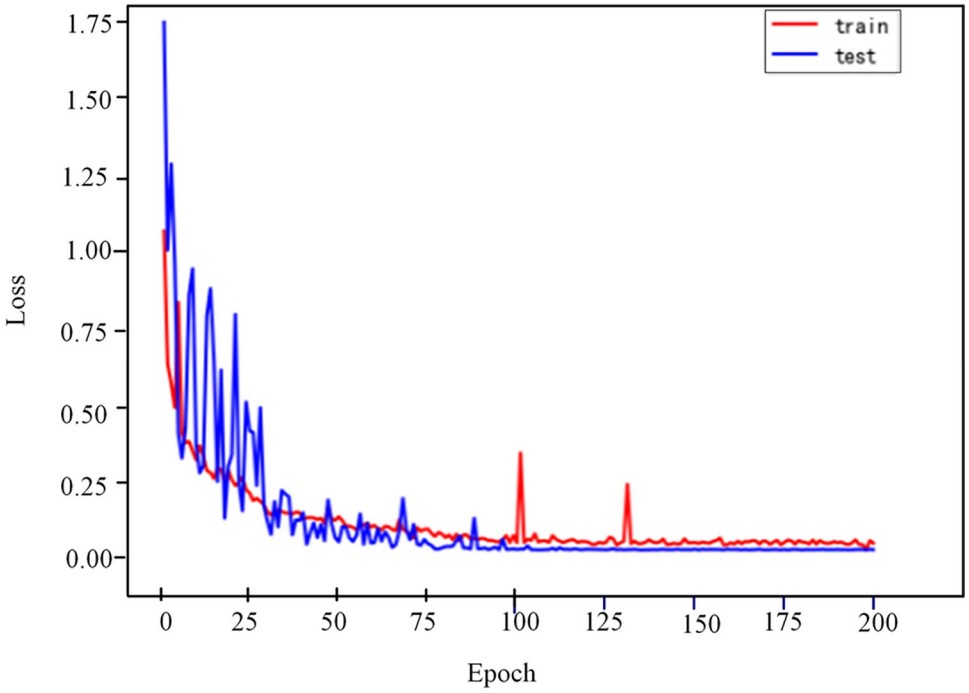

**Fig 8. Loss curve.**

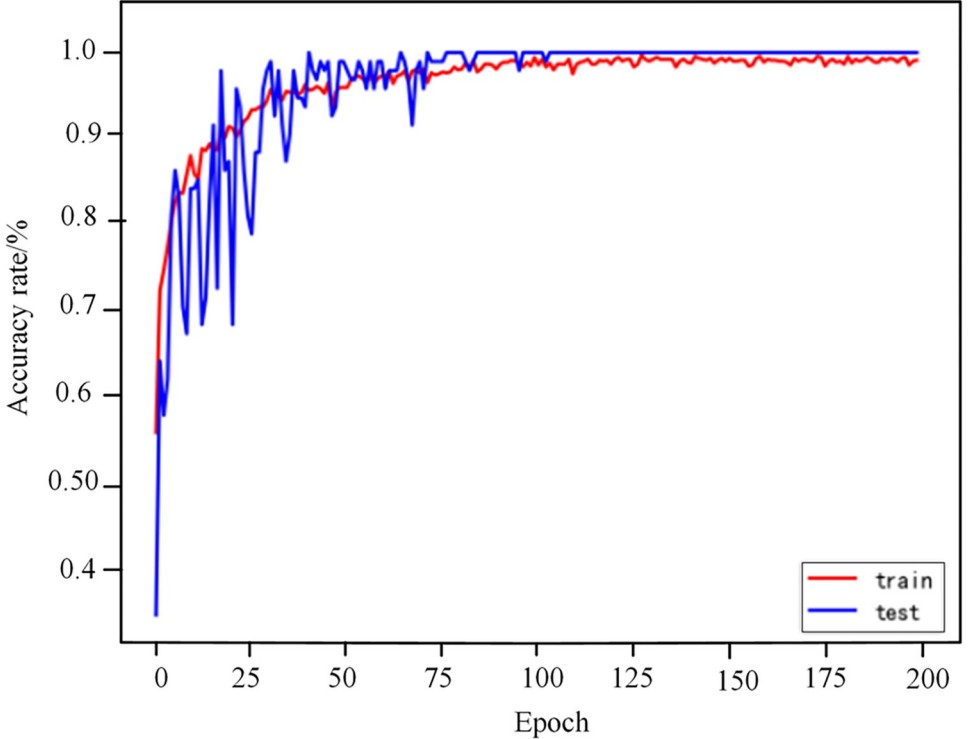

**Fig 9. Recognition accuracy rate change curve.**

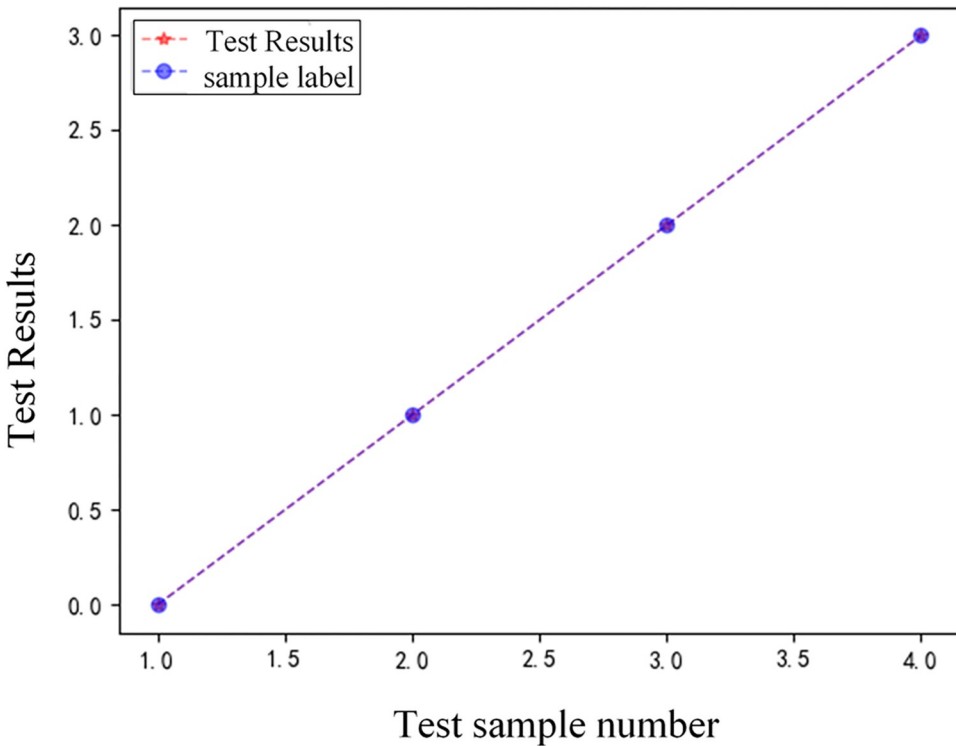

**Fig 10. Comparison of test results of different types of faults.**

GADF-VGG16 model proposed in this paper is effective and reliable for fault diagnosis of HVDC transmission lines.

## 5.3 Analysis of experimental results

**5.3.1 Sensitivity analysis.** In this section, the proposed technique performance is investigated through different fault characteristics applied along the HVDC Transmission line model presented in Fig 1.The obtained results considering the effects of the different fault resistances, fault distances and types are shown in 1), 2), 3), respectively.

1) Analysis of identification results of different types of faults

In order to verify the recognition effect of the improved VGG16 on different types of faults, taking the four fault conditions of F1~F4 in Fig 1 as an example, 4 test samples are set. And the test samples are input into the trained VGG16 for testing. The test result comparison chart is shown in Fig 10, and the corresponding fault simulation test results are shown in Table 2.

**Table 2. Simulation verification results of different types of faults.**

| Fault type | Transition resistance/Ω | Fault distance/km | Classification label | Recognition result | |
|---|---|---|---|---|---|
| | | | | Output label | Fault type |
| Positive fault in the area | 150 | 300 | 1 | 1 | PG |
| Negative fault in the area | 150 | 300 | 2 | 2 | NG |
| Bipolar fault | 150 | 300 | 3 | 3 | PNG |
| Failure occurred outside the zone | 150 | _ | 0 | 0 | EG |

**Table 3. Experimental results under different transition resistance.**

| Fault type | Transition resistance/Ω | Fault distance/km | Classification label | Recognition result | |
|---|---|---|---|---|---|
| | | | | Output label | Fault type |
| Positive fault in the area | 250 | 300 | 1 | 1 | PG |
| | 350 | | 1 | 1 | PG |
| | 650 | | 1 | 1 | PG |
| Negative fault in the area | 250 | 300 | 2 | 2 | NG |
| | 350 | | 2 | 2 | NG |
| | 650 | | 2 | 2 | NG |
| Bipolar fault | 250 | 300 | 3 | 3 | PNG |
| | 350 | | 3 | 3 | PNG |
| | 650 | | 3 | 3 | PNG |
| Failure occurred outside the zone | 250 | — | 0 | 0 | EG |
| | 350 | | 0 | 0 | EG |
| | 650 | | 0 | 0 | EG |

Analysis of Fig 10 and Table 2 shows that under the same fault distance and the same transition resistance conditions, the model proposed in this paper can accurately realize the internal and external fault detection and fault pole selection, indicating that the fault detection model is not affected by the type of HVDC transmission lines.

2) GADF-VGG16 model analysis of the results of different transition resistance fault identification

To verify the performance of the protection algorithm when different transition resistance faults occur, set four types of fault conditions at points F1 to F4 in Fig 1, the fault distance remains the same, and set 12 samples to form a test sample set. Use the model proposed in this paper to test the test samples. The experimental results are shown in Table 3. Fig 11 is a comparison diagram of the model's classification results under different transition resistance conditions.

According to analysis of Fig 11 and Table 3, the algorithm proposed in this paper can accurately identify the type of fault under the influence of different transition resistances, indicating that the model has a certain ability to withstand transition resistance.

3) GADF-VGG16 model analysis of fault recognition results under different fault distances

In order to verify the performance of the protection algorithm under different fault distances, this paper sets four fault conditions at points F2 to F4 in Fig 1. Tested in the VGG16 model, the experimental results are shown in Table 4, and the comparison of classification results under different fault distances is shown in Fig 12.

Analysis of Fig 12 and Table 4 show that under different fault distances, both near-end and far-end faults can be accurately identified, indicating that the fault detection model proposed in this paper is less affected by the fault distance.

### 5.3.2 Ablation experiment

In order to verify the effectiveness of the improved network for fault diagnosis of HVDC transmission lines, this paper uses ablation experiments to conduct longitudinal comparative analysis, and judges whether the improved algorithm is effective from the results of network recognition accuracy, parameter size and network detection speed. Table 5 shows the results of ablation experiments.

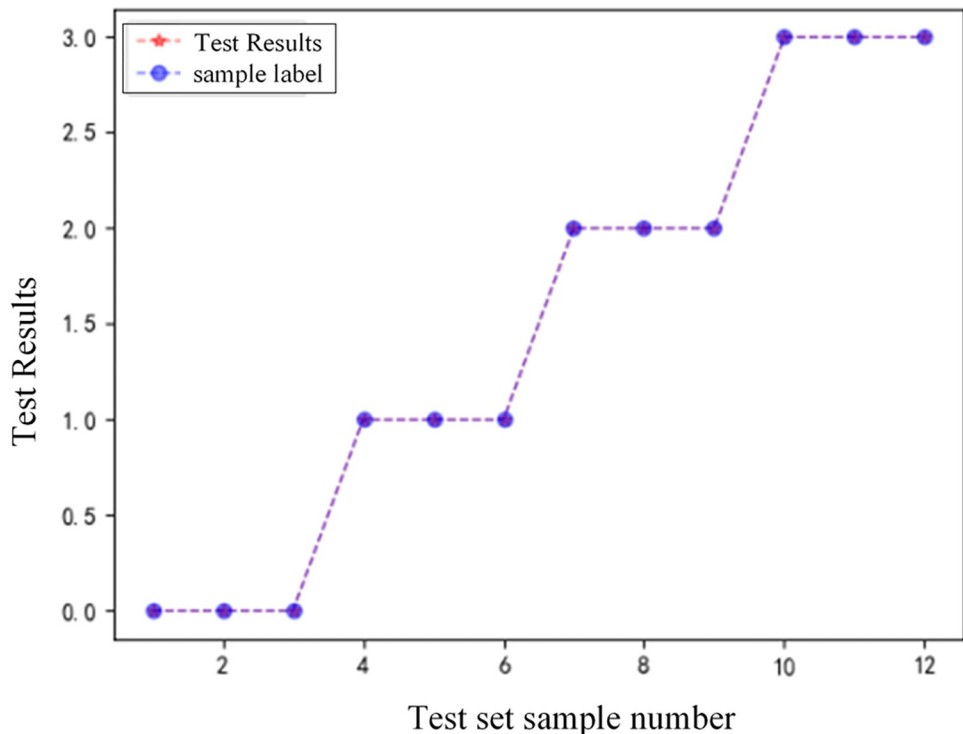

**Fig 11. Comparison chart of classification results.**

According to Table 5, after adding the BN layer to the VGG16 structure, the accuracy of the training set and the test set are increased by 0.96% and 0.61% respectively, indicating that the batch normalization process reduces the impact of the hidden layer data distribution change on the network. The dependence of the network on parameters is reduced, thereby improving the accuracy of network fault recognition; after using the dense connection structure to improve the convolutional layer, the amount of network parameters increased by 61MB, and the accuracy rate increased by 1.65%.

**Table 4. Experimental results of different fault distance.**

| Fault type | Transition resistance/Ω | Fault distance/km | Classification label | Recognition result | |
|---|---|---|---|---|---|
| | | | | Output label | Fault type |
| Positive fault in the area | 300 | 10 | 1 | 1 | PG |
| | | 100 | 1 | 1 | PG |
| | | 200 | 1 | 1 | PG |
| | | 990 | 1 | 1 | PG |
| Negative fault in the area | 300 | 150 | 2 | 2 | NG |
| | | 250 | 2 | 2 | NG |
| | | 350 | 2 | 2 | NG |
| | | 450 | 2 | 2 | NG |
| Bipolar fault | 300 | 500 | 3 | 3 | PNG |
| | | 550 | 3 | 3 | PNG |
| | | 600 | 3 | 3 | PNG |
| | | 650 | 3 | 3 | PNG |

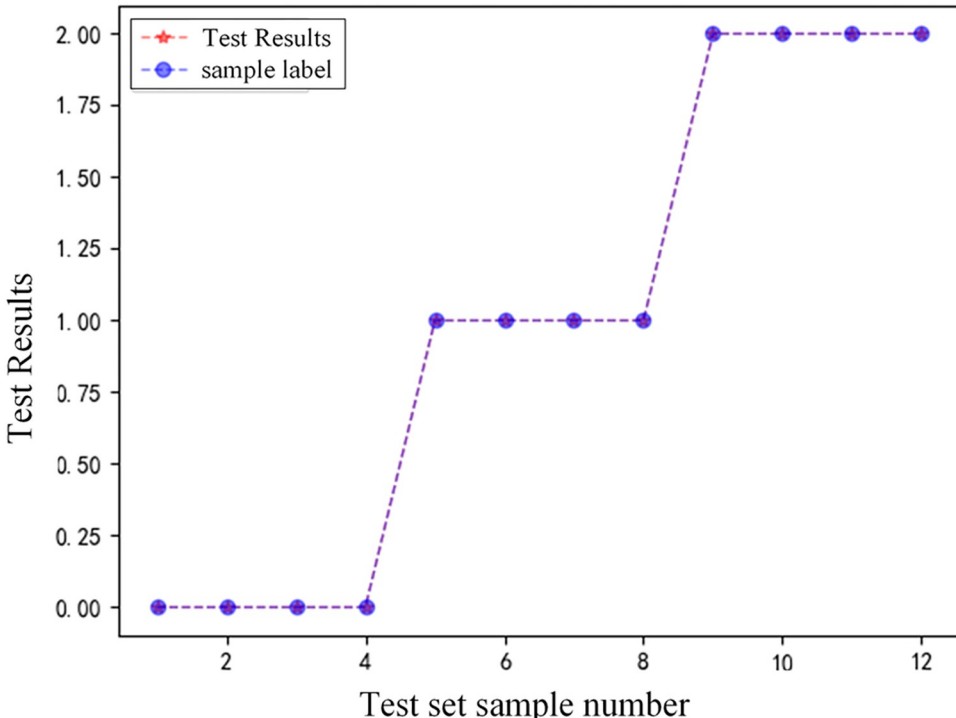

**Fig 12. Comparison chart of classification results.**

It shows that the densely connected structure can obtain richer characteristic information, and improve the network's detection effect. After using GAP instead of the fully connected layer for training, the amount of network parameters is significantly reduced by 188MB, and the detection speed (Frame Per Second, fps) is significantly improved, 31 pictures per second are detected, which is 7 more pictures per second than the detection speed of original VGG16. It shows that the global average pooling layer plays a significant role in reducing the amount of network parameters and can significantly speed up the calculation of the network. Through the longitudinal comparison of the ablation experiment, it can be seen that each improvement point has a different effect on the improvement of the network recognition accuracy, the reduction of the parameter amount and the improvement of the network detection speed. Finally, training the same data set with the new VGG16 shows that the accuracy of the network test set is increased by 2.69%, the network parameters are reduced by 179MB, and the detection speed is 26fps to meet the requirements of rapid mobility. It shows that the improvement of the network in this article is real and effective.

**Table 5. Results of ablation experiments.**

| Vgg16 | BN layer | OSA | GAP | Test set accuracy /% | Parameter /MB | detection speed(fps) |
|---|---|---|---|---|---|---|
| √ | × | × | × | 97.31 | 248 | 24 |
| √ | √ | × | × | 97.92 | 248 | 24 |
| √ | × | √ | × | 98.96 | 309 | 24 |
| √ | × | × | √ | 91.67 | 60 | 31 |
| √ | √ | √ | √ | 100 | 69 | 26 |

"√" means that the structure is added, and "×" means that the structure is not added

**Table 6. Performance analysis of protection algorithm under noise interference.**

| Fault Type | Transition resistance /Ω | Fault distance/km | Noise/dB | Category label | Identification result |
|---|---|---|---|---|---|
| | | | | | Output label |
| Positive fault in the area | 200 | 50 | 30 | PG | PG |
| | | | 20 | PG | PG |
| | | | 10 | PG | PG |
| Negative fault in the area | 300 | 950 | 30 | NG | NG |
| | | | 20 | NG | NG |
| | | | 10 | NG | NG |
| Bipolar fault | 600 | 550 | 30 | PNG | PNG |
| | | | 20 | PNG | PNG |
| | | | 10 | PNG | PNG |
| Failure occurred outside the zone | 500 | ----- | 30 | EG | EG |
| | | | 20 | EG | EG |
| | | | 10 | EG | EG |

### 5.3.3 Algorithm performance analysis.

1) Analysis of anti-noise performance

Considering that the actual HVDC transmission lines will be affected by noise interference in the event of fault, it is particularly important to verify the anti-noise performance of the algorithm. In order to simulate the noise interference in the real environment, Gaussian white noise with different signal-to-noise ratios is added to experimental test data. Signal-to-noise ratio (SNR) is defined as the ratio of signal power to noise power, the unit is dB, and is defined as follow

$$SNR = 10\lg\left(\frac{p_s}{p_n}\right) \tag{5}$$

Where $P_S$ represents signal power, and $P_n$ represents noise power.

We sets up four types of faults and three signal-to-noise ratios of 30dB, 20dB, and 10dB. A total of $4 \times 3 = 12$ samples are set, and the trained VGG16 is used for prediction. The test results are shown in Table 6.

From Table 6, it can be seen that the algorithm proposed in this paper can achieve accurate fault detection even when the SNR is 10dB. Therefore, the algorithm proposed in this paper is less affected by noise and has a certain anti-noise ability.

2) Data loss performance analysis

Since the data collection in the actual operation of the HVDC transmission line relies on the field equipment, there may be data loss in the uploaded data due to equipment failure due to environmental influences. Therefore, in order to verify the data loss resistance of the proposed algorithm, the test data are subjected to random loss processing. In this paper, four cases of faults, that is, fault occurred out of the protection zone F2, faults occurred within the protection zone F2, F3, and F4 are set, and the fault simulation is performed as shown in Fig 1. And the number of missing points is set to 15, 20, 30, 50, 75.The test sets were subjected to different levels of data loss and then converted to 2D images by GADF coding and fed into a modified VGG16 model for data loss resistance testing. The test set identification accuracy was obtained as shown in the blue curve in Fig 13. In order to further improve the model's ability to resist data loss and generalization, 100 new sets of data (The number of missing data per sample set is set to 25) are added to the original training set to form an enhanced training set. Then the

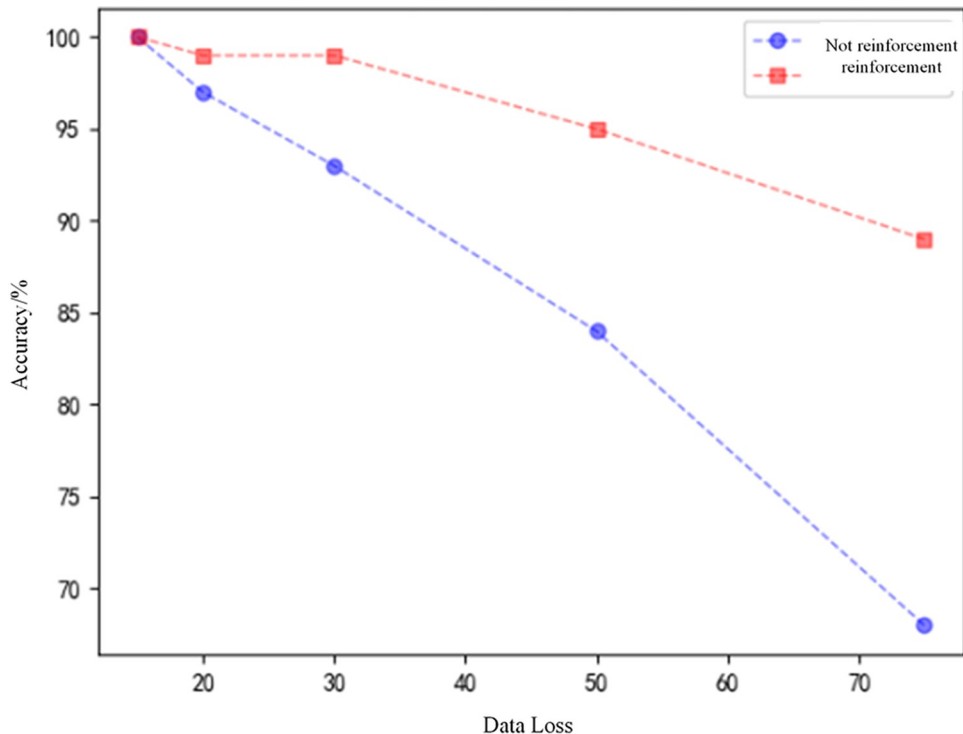

**Fig 13. Test set accuracy rate under different data loss numbers.**

improved VGG16 is trained again by using the reinforcement training set, so that the model can perform reinforcement learning on data loss features. The retrained model was then tested on the test set, which was also subjected to data loss, and the recognition accuracy of the test set is shown in the red curve in Fig 13.

Analysis of Fig 13 shows that the improved VGG16 model without reinforcement learning has lower recognition rate of the images after data loss when the number of data loss >30. It can be seen that when the data is lost to 50, the recognition accuracy is only 84%. While a small number of data loss samples were added to the training set, and after reinforcement learning, the accuracy rate reached 95% when the data loss reached 50. It can be seen that the improved VGG16 model can quickly learn the data loss features, and the model identification accuracy has been more significantly improved even in the environment with a large amount of data loss. Therefore, the algorithm proposed in this paper is less susceptible to data loss and has a certain ability to resist data loss.

**5.3.4 Comparative analysis of recognition effects of different network structures.** In order to fully verify the advantages of GADF-VGG16, we compared and verified with the VGG16 network before the improvement. At the same time, Alexnet and LeNet were added as comparative experiments, both of which were implemented using the Keras library with TensorFlow as the backend. All models use the same input pictures (GADF encoded picture), and divided into the training set and the validation set according to the ratio of 9:1 and 8:2, and each model is tested five times. The accuracy rate is the average of five experimental results, and the experimental results are shown in Table 7.

According to the results in Table 7, the improved VGG16 model proposed in this paper outperforms all other models in terms of fault identification accuracy. The accuracy of the method for fault identification is above 99%, and the test set accuracy reaches 100% when the

**Table 7. Different network algorithm fault diagnosis experiment results.**

| Training set: validation set = 9:1 | | | Training set: validation set = 8:2 | |
|---|---|---|---|---|
| Network Model | Training set accuracy/% | Testing set accuracy/% | Training set accuracy/% | Testing set accuracy/% |
| LeNet | 88.46 | 92.08 | 88.02 | 91.88 |
| Alexnet | 96.63 | 98.56 | 95.44 | 97.92 |
| Unimproved VGG16 | 97.70 | 97.31 | 95.64 | 96.98 |
| Improved GADF-VGG16 | 99.01 | 100 | 99.27 | 99.48 |

ratio of training set and validation set is 9:1.In comparison with lenet and alexnet, it can be seen that the deep network model VGG16 is better at recognition because the deep network model can better learn the deep abstract features of the fault data.Comparing the improved network model in this paper with the unimproved VGG16 model, it can be seen that the recognition accuracy of the improved VGG16 model is further improved.

It shows that the GADF-VGG16 HVDC transmission line fault diagnosis model proposed in this paper can identify HVDC transmission line faults more accurately and efficiently.

**5.3.5 Comparison with other intelligent fault diagnosis methods.** In order to further validate the effectiveness of the proposed method in this paper, it is compared with existing neural network models in intelligent fault diagnosis methods such as SVM, BP,ANN, RF, CNN. The obtained test results are shown in Table 8. It can be seen from Table 8 that the model proposed in this paper has the highest identification rate of transmission line faults among the six models. It can be seen that the model proposed in this paper has a better identification effect and can effectively solve the problem of transmission line fault identification.

**5.3.6 Algorithm calculation complexity.** The pros and cons of an algorithm are mainly measured from two aspects: the execution time of the algorithm (time complexity) and the required storage space (space complexity).

1. The time complexity of the VMD algorithm can be expressed as: *Time*~O(n), where n denotes the number of program executions, and in this paper n = k, k denotes the number of IMF components of the vmd decomposition.

2. The time complexity of the GADF algorithm can be expressed as: *Time*~O(n), where n is the number of times the program is executed, and in this paper n is equal to the number of samples.

3. Time complexity of the convolutional neural network VGG16:

In a convolutional neural network, the computing time is mainly spent on the convolution operation and the operation of the fully connected layer, and the operation of the fully connected layer accounts for most of the time. The greater the number of operations, the greater the computational resources consumed and the longer the time spent. In this paper, in order

**Table 8. Model recognition accuracy comparison.**

| Neural network | training set accuracy/% | test set accuracy/% |
|---|---|---|
| VGG16 | 99.01 | 100 |
| SVM | 92.84 | 92.55 |
| BP | 88.0 | 84.3 |
| ANN | 86.66 | 80.65 |
| RF | 98.55 | 98 |
| CNN | 88.46 | 92.08 |

to reduce the time complexity of VGG16, the global average pooling layer is used instead of the fully connected layer, which greatly reduces the amount of computation in the network, i.e. greatly reduces the time complexity of the network.

The time complexity of a single convolutional layer is defined as (6):

$$Time \sim O(M^2 \cdot k^2 \cdot C_{in} \cdot C_{out}) \tag{6}$$

$$M = (X - K + 2 * Padding)/Stride + 1 \tag{7}$$

Where M represents the size of the output feature map of each convolution kernel, and its expression is as in Eq (7), which is determined by the four parameters of input matrix size X, convolution kernel size K, padding, and stride; In the Eq (6), k represents the edge length of each convolutional kernel Kernel; $C_{in}$ represents the number of channels of each convolution kernel, that is, the number of input channels; and $C_{out}$ represents the number of convolutional kernels, that is, the number of output channels. It can be seen that the time complexity of each convolutional layer is fully determined by the output feature area $M^2$ the convolutional kernel area $k^2$, and the number of input and output channels.

The overall time complexity of the VGG16 network can be expressed as (8):

$$Time \sim O(\sum_{l=1}^{D} M_l^2 \cdot k_l^2 \cdot C_{l-1} \cdot C_l + \sum_{j=1}^{D_{FC}} C_{j-1} \cdot M_j^2 \cdot m_j) \tag{8}$$

Where D represents the number of all convolutional layers of the neural network, that is, the network depth; l represents the first convolutional layer of the neural network; $C_l$ represents the number of output channels $C_{out}$ of the lth convolutional layer of the neural network, that is, the number of convolutional kernels in that layer. $D_{FC}$ represents the number of fully connected layers, and $m_j$ represents the number of features to be output in that layer.

In this paper, the global average pooling layer is used to replace the fully connected layer, so the overall time complexity of the VGG16 network proposed in this paper can be expressed as (9):

$$Time \sim O(\sum_{l=1}^{D} M_l^2 \cdot k_l^2 \cdot C_{l-1} \cdot C_l + C_{l-1} \cdot m_j) \tag{9}$$

4) Space Complexity of convolutional neural network VGG16:The space complexity is a measure of the size of the storage space temporarily occupied by an algorithm during the running process, which can be expressed as (10):

$$Space \sim O(\sum_{l=1}^{D} K_l^2 \cdot C_{l-1} \cdot C_l + \sum_{l=1}^{D} M^2 \cdot C_l) \tag{10}$$

In the ablation experiment of 4.3.2, it can be seen that the size of the network parameters of the improved VGG16 algorithm proposed in this paper is 69MB after removing the fully connected layer. Compared with the unimproved VGG16, it can be seen that the memory occupied by the network parameters is reduced by 179MB. It can be seen that the improved VGG16 has greatly improved the space complexity.

## 6 Conclusion

Aiming at the problem that the existing fault diagnosis methods are difficult to accurately extract fault characteristics when faced with complex fault characteristics, this paper proposes a fault diagnosis method for HVDC transmission lines based on GADF-VGG16. The algorithm proposed in this paper breaks the traditional thinking, converts the one-dimensional fault signal into a two-dimensional image for self-adaptive extraction of deep fault features, and better combines the advantages of convolutional neural network in image processing to realize intelligent fault diagnosis of HVDC transmission lines. The simulation results show that the proposed algorithm has the following advantages:

The method proposed in this paper uses GADF to convert one-dimensional fault data into two-dimensional color images, retains the temporal correlation in the data, and uses deep learning algorithms to achieve high-dimensional feature extraction. Compared with the method of converting the data into a two-dimensional grayscale image in the literature [18], the data processing method proposed in this paper will not cause the loss of characteristic data, so that the fault diagnosis model has a higher fault identification accuracy.

In this paper, the VGG16 network is improved by adding BN layer and convolutional layer of densely connected structure to speed up the training and convergence speed of the network while realizing feature reuse and enhancement; at the same time, the global average pooling layer is used to replace the fully connected layer, reducing The amount of model parameters and computation time make the proposed method more suitable for rapid fault diagnosis.

Compared with traditional intelligent algorithms, the fault diagnosis model based on improved VGG16 constructed in this paper can fully extract deep fault features. Compared with the fault diagnosis algorithm based on SVM, BP, RF and other shallow neural networks in literature [9–12], the algorithm proposed in this paper has higher fault recognition accuracy, and the experimental results show that it is not affected by transition resistance, fault type and fault. It has strong anti-interference ability and fault tolerance.

On the other hand, the method proposed in this paper also has some limitations. Due to the huge size of the actual HVDC transmission system, the fault situation of the transmission line is more complicated. Limited to laboratory conditions, the fault diagnosis models proposed in this paper are all carried out in a simulation environment, and there is a lack of fault data in the actual operation process to verify the proposed model.

In order to further improve the fault diagnosis model of HVDC transmission lines, how to verify the proposed diagnosis model in combination with the actual fault data of HVDC transmission lines is the focus of the next research work.

## Supporting information

**S1 File. Author information.**
(DOCX)

**S1 Dataset. Train set.**
(XLSX)

**S2 Dataset. Test set.**
(XLSX)

**S3 Dataset. Test sample data.**
(XLSX)

**S4 Dataset. Data loss.**
(XLSX)

**S5 Dataset. Different SNR.**
(XLSX)

## Acknowledgments

Conceived and designed the experiments: HAO WU. Performed the experiments: Yuping Yang. Analysed the data: Yuping Yang, Sijing Deng, Qiaomei Wang. Contributed reagents/materials/analysis tools: HAO WU, Sijing Deng, Qiaomei WANG, Hong Song. Wrote the paper: Yuping Yang.

## Author Contributions

**Data curation:** Yuping Yang, Sijing Deng, Qiaomei Wang.

**Project administration:** Hao Wu.

**Supervision:** Hao Wu, Hong Song.

**Validation:** Hao Wu, Sijing Deng, Qiaomei Wang.

**Writing – original draft:** Yuping Yang.

**Writing – review & editing:** Yuping Yang.

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
