## [Decision Letter · Decision Letter 0]

3 May 2022

PONE-D-22-08053GADF-VGG16 based Fault diagnosis method for HVDC transmission linesPLOS ONE

Dear Dr. Yang,

Thank you for submitting your manuscript to PLOS ONE. After careful consideration, we feel that it has merit but does not fully meet PLOS ONE’s publication criteria as it currently stands. Therefore, we invite you to submit a revised version of the manuscript that addresses the points raised during the review process.

We look forward to receiving your revised manuscript.

Kind regards,

Ashwani Kumar, Ph.D.

Academic Editor

PLOS ONE

Journal Requirements:

2. Please ensure you have included in the Methods section of your manuscript information regarding the origin of the dataset used in your study. If the supporting information files include the raw data used in this study, please ensure you state clearly in your manuscript text which dataset is used in each experiment.

Please also ensure that Table 6 is translated into English.

6. Please upload a new copy of Figures as the details are not clear. Please follow the link for more information: " ext-link-type="uri" xlink:type="simple">https://blogs.plos.org/plos/2019/06/looking-good-tips-for-creating-your-plos-figures-graphics/"
" ext-link-type="uri" xlink:type="simple">https://blogs.plos.org/plos/2019/06/looking-good-tips-for-creating-your-plos-figures-graphics/"

Reviewers' comments:

Reviewer's Responses to Questions

**Comments to the Author**

1. Is the manuscript technically sound, and do the data support the conclusions?

Reviewer #1: Yes

Reviewer #2: Partly

2. Has the statistical analysis been performed appropriately and rigorously? 

Reviewer #1: Yes

Reviewer #2: No

3. Have the authors made all data underlying the findings in their manuscript fully available?

Reviewer #1: Yes

Reviewer #2: Yes

4. Is the manuscript presented in an intelligible fashion and written in standard English?

Reviewer #1: Yes

Reviewer #2: Yes

5. Review Comments to the Author

Reviewer #1: 1) The core technologies lack original innovation.

2) The literature review is problematic. The review on technical development is insufficient.For example, https://doi.org/10.3390/app12063139；
https://doi.org/10.1109/TIM.2022.3159005;
https://doi.org/10.2147/DMSO.S341364；https://doi.org/10.1016/j.ins.2021.11.052;
https://doi.org/10.1109/ACCESS.2021.3108972 and so on

3) The contributions should be simplified to highlight your work.

4) The authors spent too many pages introducing some well-known theories. It can be simplified.

5) The effectiveness of the proposed method needs to be verified by contrast experiments.

6) The comparison results with existing works should be proved by experiments rather than written description.

Reviewer #2: This paper presents "GADF-VGG16 based Fault diagnosis method for HVDC transmission lines" . However, this paper can not be accept in the present format.

1. Improve the writing style of the paper. Like, avoid writing like Reference [8] shows that....

2. Add the author lists in the paper and same in the submission data.

3. Improve the quality of the Images. It is hard to understand the meanings.

4. Improve the results and discussion section by improving the comparison matrices like computational time and implications implementations.

5. Add sensitivity analysis with respect the data uncertainty.

6. PLOS authors have the option to publish the peer review history of their article (what does this mean?). If published, this will include your full peer review and any attached files.

Reviewer #1: No

Reviewer #2: No

---

## [Author Response · Author response to Decision Letter 0]

20 Jun 2022

Reviewer #1:

Comment:

1） The literature review is problematic. The review on technical development is insufficient.

2) The contributions should be simplified to highlight your work.

3) The authors spent too many pages introducing some well-known theories. It can be simplified.

4）The effectiveness of the proposed method needs to be verified by contrast experiments.

5) The comparison results with existing works should be proved by experiments rather than written description.

An explanation of modification:

1、In response to question 1)，The authors supplemented the literature review section as follows:

At present, experts and scholars have proposed a series of protection schemes for the protection of HVDC transmission lines. Among them, traveling wave protection, differential undervoltage protection is generally used as the main protection, longitudinal differential protection and low voltage protection as backup protection [4], while the traveling wave protection principle is most widely used in many protection methods. For example, in [5], based on the double-ended traveling wave method, it is proposed to use the time interval between the detection of the first incident traveling wave and the reflected wave at the detection point for the identification of faults inside and outside the zone. However, the method relies on the transmission of double-ended information, which has an impact on the quickness of the protection. Considering that traveling wave protection is susceptible to high resistance and fault distance, the author of [6] proposes to use traveling wave transmission principle combined with Teager energy operator to differentiate the internal fault from the external fault in the HVDC system. In this method, fault identification can be completed in a short time and the identification of fault types can be accurately achieved under high resistance faults at the remote end.However, how to accurately identify the traveling wave head in the traveling wave method is an insurmountable technical problem [7]. In [8],studies the internal and external faults identify method of HVDC transmission system based on distributed parameter model. This method does not rely on the traveling wave protection principle, and the accuracy of its fault detection depends on the setting of parameters of the transmission line.

In recent years, a large number of scholars have proposed the use of support vector machines (SVM), Back propagation neural networks（BP）, artificial neural networks (ANN), random forests and other methods [9-12] to study the problem of transmission line fault diagnosis.For example, Johnson et al [9] used the SVM classification mechanism to achieve fault identification and classification of HVDC transmission lines.The feature vector used in the classification modules is the standard deviation of the signals over half cycle before and after the occurrence of fault.However, the method does not fully exploit the waveform characteristics of the faulty traveling waves and the fault tolerance of the method needs to be further verified.

R. Kou et al [10] used the amount of electrical variation in transmission line fault conditions as a feature vector to train BP neural networks for fault diagnosis.This method has a good identification effect for in-zone faults, but does not consider the identification of out-of-zone faults.K. Moloi et al [11] used a particle swarm algorithm (POS) to optimize an artificial neural network (ANN), and then used the optimized ANN model to identify and classify different faults.The method can identify faults in and out of the zone with an accuracy of 99%. However, whether the method can also correctly identify fault types in high transition resistance and high noise environments remains to be proven.Wu et al [12] used random forest (RF) neural networks to achieve the selection of fault poles and identification of fault types on HVDC transmission lines. But, the extraction of fault features is more complex with this method.

Yet, the development of deep learning has brought new ideas to the field of transmission line fault diagnosis.The deep learning approach is able to learn deep abstract features in the data autonomously [13] and is suitable for complex transmission line fault diagnosis problems.Zhai et al [14] proposed to construct a fault diagnosis model for HVDC transmission lines using an improved convolutional neural network (CNN) to extract features and implement fault classification for current timing data.Compared with the traditional CNN network, the method has a certain improvement in recognition accuracy, but the CNN is not ideal for feature extraction of time-series signals.Therefore, it has been proposed to convert the time-series signal into a two-dimensional image, and then use CNN for fault feature extraction and classification.As Wang et al [15] proposed to convert one-dimensional time-series signals into two-dimensional grayscale images, and then use CNN for transmission line fault classification.However, the processing of the time-series signal into a grey-scale image in this method results in the loss of feature data.It can be seen that the application of deep learning to fault diagnosis on transmission lines is feasible and is subject to further in-depth research.

2、In response to question 2) and 3)，the author have simplified the paper,it can be seen in the revised manuscript.

3、In response to question 4）and 5），the author added the contrast experiment as follow:

It is compared with the published work related to intelligent fault diagnosis of HVDC transmission lines [9-14].In order to objectively verify the effectiveness of the method proposed in this paper, it is compared with the neural network models in the above intelligent fault diagnosis methods such as SVM, BP, ANN, RF, and CNN.The obtained test results are shown in Table 8.It can be seen from Table 8 that the model proposed in this paper has the highest recognition rate of transmission line faults among the six models.It can be seen that the model proposed in this paper has a better identification effect and can effectively solve the problem of transmission line fault identification.

Table 8 Model recognition accuracy comparison

Neural network training set accuracy/% test set accuracy/%

VGG16 99.01 100

SVM 92.84 92.55

BP 88.0 84.3

ANN 86.66 80.65

RF 98.55 98

CNN 88.46 92.08

Reviewer #2:

Comment:

1)Improve the writing style of the paper. Like, avoid writing like Reference [8] shows that....

2) Improve the quality of the Images. It is hard to understand the meanings.

3) Improve the results and discussion section by improving the comparison matrices like computational time and implications implementations.

4)Add sensitivity analysis with respect the data uncertainty.

An explanation of modification:

1、In response to question 1）、2）：

The writing style of the paper and the images has been modified as required.

2、In response to question 3): 

In the results analysis section, the advantages of the proposed algorithm are comprehensively demonstrated from five aspects: data sensitivity analysis, ablation experiment analysis, algorithm performance analysis, comparison analysis of the recognition effect of different network structures, and comparison analysis with existing intelligent diagnosis algorithms.

①Sensitivity analysis:In this section, the proposed technique performance is investigated through different fault characteristics applied along the HVDC Transmission line model presented in Fig.1. Experimental validation using the control variable method with different influencing factors such as different fault types, different transition resistances and different fault distances .It demonstrates that the fault diagnosis model proposed in this paper is not affected by the fault type. It also demonstrates that the model is less affected by the fault distance, and has a certain ability to withstand transition resistance. 

②In the ablation experiment: In this section, the ablation experiments have verified that all the improvement points of this paper for VGG16 are effective.By the way, the FPS index is mentioned in Table 5 , that is the detection speed of the network. The improved VGG16 detection speed can detect 26 pictures per second, that is the algorithm proposed in this paper needs 0.038s to diagnose a fault.It meets the requirements of rapid mobility.

③Algorithm performance analysis: In this section, the performance of the fault diagnosis algorithm proposed in this paper is analyzed, and its anti-noise performance and anti-data loss performance are verified respectively. In addition, the proposed approach is robust to measurement noise and data loss errors.

④Comparative analysis of recognition effects of different network structures：In this section, We compare the network model proposed in this paper with the unimproved VGG16 network model as well as the LeNet and Alexnet network models。In comparison with lenet and alexnet, it can be seen that the deep network model VGG16 is better at recognition because the deep network model can better learn the deep abstract features of the fault data.Comparing the improved network model in this paper with the unimproved VGG16 model, it can be seen that the recognition accuracy of the improved VGG16 model is further improved.It shows that the GADF-VGG16 HVDC transmission line fault diagnosis model proposed in this paper can identify HVDC transmission line faults more accurately and efficiently.

⑤Comparison with other intelligent fault diagnosis methods:It is compared with the published work related to intelligent fault diagnosis of HVDC transmission lines [9-14].In order to objectively verify the effectiveness of the method proposed in this paper, it is compared with the neural network models in the above intelligent fault diagnosis methods such as SVM, BP, ANN, RF, and CNN.The result shows that the model proposed in this paper has the highest recognition rate of transmission line faults among the six models.It can be seen that the model proposed in this paper has a better identification effect and can effectively solve the problem of transmission line fault identification.

3、In response to question 4):

 Sensitivity analysis of data uncertainty can be found in Section 4.3.1 of this article.In this section, the proposed technique performance is investigated through different fault characteristics applied along the HVDC Transmission line model presented in Fig.1.The obtained results considering the effects of the different fault resistances,fault distances and types are shown in 1)、2)、3) , respectively.

---

## [Decision Letter · Decision Letter 1]

6 Jul 2022

PONE-D-22-08053R1GADF-VGG16 based Fault diagnosis method for HVDC transmission linesPLOS ONE

Dear Dr. wu,

Thank you for submitting your manuscript to PLOS ONE. After careful consideration, we feel that it has merit but does not fully meet PLOS ONE’s publication criteria as it currently stands. Therefore, we invite you to submit a revised version of the manuscript that addresses the points raised during the review process.Please submit your revised manuscript by Aug 20 2022 11:59PM. If you will need more time than this to complete your revisions, please reply to this message or contact the journal office at plosone@plos.org. Please include the following items when submitting your revised manuscript:A rebuttal letter that responds to each point raised by the academic editor and reviewer(s). You should upload this letter as a separate file labeled 'Response to Reviewers'.A marked-up copy of your manuscript that highlights changes made to the original version. You should upload this as a separate file labeled 'Revised Manuscript with Track Changes'.An unmarked version of your revised paper without tracked changes. You should upload this as a separate file labeled 'Manuscript'.If applicable, we recommend that you deposit your laboratory protocols in protocols.io to enhance the reproducibility of your results. Protocols.io assigns your protocol its own identifier (DOI) so that it can be cited independently in the future. For instructions see: https://journals.plos.org/plosone/s/submission-guidelines#loc-laboratory-protocols. Additionally, PLOS ONE offers an option for publishing peer-reviewed Lab Protocol articles, which describe protocols hosted on protocols.io. Read more information on sharing protocols at https://plos.org/protocols?utm_medium=editorial-emailutm_source=authorlettersutm_campaign=protocols.

We look forward to receiving your revised manuscript.

Kind regards,

Ashwani Kumar, Ph.D.

Academic Editor

PLOS ONE

Journal Requirements:

Reviewers' comments:

Reviewer's Responses to Questions

**Comments to the Author**

1. If the authors have adequately addressed your comments raised in a previous round of review and you feel that this manuscript is now acceptable for publication, you may indicate that here to bypass the “Comments to the Author” section, enter your conflict of interest statement in the “Confidential to Editor” section, and submit your "Accept" recommendation.

Reviewer #1: (No Response)

Reviewer #3: All comments have been addressed

2. Is the manuscript technically sound, and do the data support the conclusions?

Reviewer #1: No

Reviewer #3: Yes

3. Has the statistical analysis been performed appropriately and rigorously? 

Reviewer #1: N/A

Reviewer #3: Yes

4. Have the authors made all data underlying the findings in their manuscript fully available?

Reviewer #1: (No Response)

Reviewer #3: Yes

5. Is the manuscript presented in an intelligible fashion and written in standard English?

Reviewer #1: (No Response)

Reviewer #3: Yes

6. Review Comments to the Author

Reviewer #1: According to the revised paper, I have appreciated the deep revision of the contents and the present form of this manuscript. There is little content, which need be revised according to the comment of reviewer in order to meet the requirements of publish. A number of concerns listed as follows:

(1) The authors need to interpret the meanings of the variables.

(2) Please highlight your contributions in introduction.

(3) The abstract should be rewritten to reflect the significance of the proposed work. The current abstract shows a lot of background information.

(4) Conclusion: What are the advantages and disadvantages of this study compared to the existing studies in this area?

(5) The inspiration of your work must further be highlighted. Some suggested recent literatures should add. For example, Zhou, X.B, Ma, H.J., Gu J.G., Chen, H.L., Deng, W. Parameter adaptation-based ant colony optimization with dynamic hybrid mechanism. Eng. Appl. Artif. Intel. 2022, https://doi.org/10.1016/j.engappai.2022.105139；https://doi.org/10.1109/JSEN.2022.3179165；
https://doi.org/10.1109/TR.2022.3180273；
https://doi.org/10.1016/j.ins.2021.11.052 and so on and so on.

(6) Correct typological mistakes and mathematical errors.

Reviewer #3: The authors improved their paper by adding important parts making their research results more clear to the readers. I appreciate that. The intelligent fault detection is a more and more the subject of the research the authors trying to find better solutions for the high-voltage direct current transmission system. By the responses to the reviewers the authors clarified all the parts of their research. In my opinion this is a good paper.

7. PLOS authors have the option to publish the peer review history of their article (what does this mean?). If published, this will include your full peer review and any attached files.

Reviewer #1: No

Reviewer #3: No

---

## [Author Response · Author response to Decision Letter 1]

15 Jul 2022

Reviewer #1:

Comment 1:

(1) The authors need to interpret the meanings of the variables.

(2) Please highlight your contributions in introduction.

(3) The abstract should be rewritten to reflect the significance of the proposed work. The current abstract shows a lot of background information.

(4) Conclusion: What are the advantages and disadvantages of this study compared to the existing studies in this area?

(5) The inspiration of your work must further be highlighted. Some suggested recent literatures should add. 

An explanation of modification:

1、In response to question 1）： 

 The author adds corresponding explanations to the meanings of the variables involved in the article.The variables in the formula have been explained below the public announcement, and the variables in the table, such as EG, PG, etc., are explained in Section 1.1 of the article.

2、In response to question 2）：

This paper adopts the method of deep learning and proposes a fault diagnosis model for HVDC transmission based on the improved VGG16 network. The main contributions are as follows:

1）This paper uses VMD to decompose the fault voltage signal of the HVDC transmission line into modal components, and converts the selected IMF component into a color image through the Gramian Angular Difference Field (GADF), then input the images into the improved VGG16 for feature extraction and classification. The method uses a novel GADF encoding approach for data pre-processing, constructing bijective mappings in one-dimensional time series and two-dimensional spatial series, which will not cause the loss of feature information[16].

2）This paper improves the traditional VGG16 model structure by adding the BN layer and the convolution layer with the densely connected structure to accelerate the training and convergence of the network and realize the reuse and enhancement of features. At the same time, the global average pooling layer is used to replace the fully connected layer, which reduces the amount of model parameters and calculation time, making the proposed method more suitable for rapid fault diagnosis. 

3）In this paper, the improved VGG16 model is applied to the fault diagnosis of HVDC transmission lines, and compared with other intelligent fault diagnosis methods, the results show that fault diagnosis algorithm based on GADF-VGG16 is more reliable and the fault identification accuracy is higher.

3、In response to question 3）：

 The summary section has been modified as requested as follows：

Since HVDC transmission lines undertake the important task of electric energy transportation, in order to improve the reliability of power supply and maintain the safe and stable operation of the power system, the role of fault diagnosis of transmission lines is very important.In this paper,an intelligent fault detection scheme for high-voltage direct current (HVDC) transmission line based on the Gramian Angular Difference Field (GADF) and the improved convolutional neural network VGG16(VGG16) is proposed. This method first performs variational modal decomposition (VMD) on the original fault voltage signal, and then uses the correlation coefficient method to select the appropriate intrinsic mode function (IMF) component, and converts it into a two-dimensional image using the Gramian Angular Difference Field(GADF). Finally, the improved VGG16 network is used to extract and classify fault features adaptively to realize fault diagnosis. The result of the comparative experiment show that the model proposed in this paper can adaptively extract and classify fault features, has high fault detection accuracy, and has good anti-noise ability and fault tolerance.At the same time, it is proved that the GADF-VGG16 fault identification model can be effectively applied to the fault diagnosis of transmission lines.And it also provides a new idea for further applying deep learning to the research of transmission line fault diagnosis.

4、In response to question 4）：

 Compared with the existing related research, the advantages of the method proposed in this paper are as follows:：First of all, the algorithm proposed in this paper breaks the traditional thinking, converts one-dimensional fault signals into two-dimensional images for adaptive extraction of fault features, and better combines the advantages of convolutional neural networks in image processing to realize intelligent fault diagnosis of HVDC transmission lines. Secondly, compared with other intelligent algorithms, the improved VGG16 fault identification model proposed in this paper has higher fault identification accuracy. At the same time, the algorithm proposed in this paper is less affected by noise, has strong anti-interference ability, and has strong learning ability and generalization performance.

 The disadvantages of the method proposed in this paper are as follows：Limited to laboratory conditions, the fault diagnosis models proposed in this paper are all carried out in a simulation environment, and there is a lack of fault data in the actual operation process to verify the proposed model.

5、In response to question 5）：

 To further highlight the source of inspiration for this paper, the author adds some recent literature such as [17] [22] [23] [25]. The idea of processing the original data can be seen from the literature [17], the idea of converting one-dimensional data into a two-dimensional image is mainly from the literature [22] [23], and finally the improvement idea of the VGG16 network mainly comes from Literature [25].

Reviewer #3:

Comment :

(6)Correct typological mistakes and mathematical errors：The authors improved their paper by adding important parts making their research results more clear to the readers. I appreciate that. The intelligent fault detection is a more and more the subject of the research the authors trying to find better solutions for the high-voltage direct current transmission system. By the responses to the reviewers the authors clarified all the parts of their research. In my opinion this is a good paper.

An explanation of modification:

 I am very grateful to the reviewers for their recognition. The authors have made further revisions for the shortcomings of the article. I hope that the reviewers will criticize and correct the revised manuscript submitted by the authors.

Journal Requirements:

modification to references:

Compared with the original manuscript, the added references are mainly to show some recent problems in the field of transmission line fault diagnosis research and some new research ideas. Added the following references:

[1] Li Zhenxing, Tan Hong, Ye Shiyun, Li Zhenhua, Xu Yanchun. Analysis of the research status of UHV DC line protection[J]. High-voltage electrical appliances,2018,54(05):184-189.DOI:10.13296/j.1001-1609.hva.2018.05.029.

[2]Hao, W., Mirsaeidi, S., Kang, X., Dong, X., Tzelepis, D. (2018). A novel traveling-wave-based protection scheme for LCC-HVDC systems using Teager energy operator. International Journal of Electrical Power Energy Systems, 99, 474–480.

[3]Muniappan, Mohan. (2021). A comprehensive review of DC fault protection methods in HVDC transmission systems. Protection and Control of Modern Power Systems. 6. 10.1186/s41601-020-00173-9. 

[4]Wu H, Wang Q, Yu K, Hu X, Ran M (2020) A novel intelligent fault identification method based on random forests for HVDC transmission lines. PLoS ONE 15(3): e0230717. https://doi.org/10.1371/journal.pone.0230717

[5]ZHAO Zhongqiu, ZHENG Peng, XU Shoutao, et al. Object detection with deep learning: a review [ J]. IEEE T ransactions on Neural Networks and Learning Systems, 2019, 30(11) :3212-323

[6]Wang, J.; Zheng, X.D.; Tai, N.L. DC Fault Detection and Classification Approach of MMC-HVDC Based on Convolutional Neural Network. In Proceedings of the 2018 2nd IEEE Conference on Energy Internet and Energy System Integration (EI2), Beijing, China, 20–22 October 2018. 

[7]Z. Wang, T. Oates. Encoding time series as images for visual inspection and classification using tiled convolutional neural networks[C ], Proceedings of the 2015 Association for the Advancement of Artificial Intelligence (AAAI) Workshops, pp.40- 46.

[8]Y. Shen, W. Zheng, W. Yin, A. Xu and H. Zhu, "Feature Extraction Algorithm Using a Correlation Coefficient Combined With the VMD and Its Application to the GPS and GRACE," in IEEE Access, vol. 9, pp. 17507-17519, 2021, doi: 10.1109/ACCESS.2021.3049118.

[9]Xiao Xiong, Xiao Yuxiong, Zhang Yongjun, Song Guoming, Zhang Fei.Application Research of Data Augmentation Method Based on Two-dimensional Grayscale Image in Motor Bearing Fault Diagnosis[J].Chinese Journal of Electrical Engineering,2021,41(02):738-749 .DOI: 10.13334/j.0258-8013.pcsee.200834.

[10]TONG yu, PANG xinyu, WEI zihan. Fault diagnosis method of rolling bearing based on GADF-CNN[J]. Vibration and shock, 2021,40(05):247-253+260.

[11]Yang, H., Ni, J., Gao, J. et al. A novel method for peanut variety identification and classification by Improved VGG16. Sci Rep 11, 15756 (2021). https://doi.org/10.1038/s41598-021-95240-y

At the same time, compared with the original manuscript, the author has deleted the content of the article, and the content of these references has been deleted, so the author deleted the corresponding references:

[1]Y. Lecun, L. Bottou, Y. Bengio and P. Haffner, "Gradient-based learning applied to document recognition," in Proceedings of the IEEE, vol. 86, no. 11, pp. 2278-2324, Nov. 1998, doi: 10.1109/5.726791.

[2]Krizhevsky A, Sutskever I, Hinton G E. Imagenet classification with deep convolutional neural networks[C]//Advances in neural information processing systems. 2012: 1097-1105.

[3]Simonyan K, Zisserman A. Very Deep Convolutional Networks for Large-Scale Image Recognition[J]. arXiv preprint arXiv:1409.1556, 2014

---

## [Decision Letter · Decision Letter 2]

25 Jul 2022

PONE-D-22-08053R2GADF-VGG16 based Fault diagnosis method for HVDC transmission linesPLOS ONE

Dear Dr. wu,

Thank you for submitting your manuscript to PLOS ONE. After careful consideration, we feel that it has merit but does not fully meet PLOS ONE’s publication criteria as it currently stands. Therefore, we invite you to submit a revised version of the manuscript that addresses the points raised during the review process.

We look forward to receiving your revised manuscript.

Kind regards,

Ashwani Kumar, Ph.D.

Academic Editor

PLOS ONE

Journal Requirements:

Reviewers' comments:

Reviewer's Responses to Questions

**Comments to the Author**

1. If the authors have adequately addressed your comments raised in a previous round of review and you feel that this manuscript is now acceptable for publication, you may indicate that here to bypass the “Comments to the Author” section, enter your conflict of interest statement in the “Confidential to Editor” section, and submit your "Accept" recommendation.

Reviewer #1: (No Response)

Reviewer #3: All comments have been addressed

2. Is the manuscript technically sound, and do the data support the conclusions?

Reviewer #1: Yes

Reviewer #3: Partly

3. Has the statistical analysis been performed appropriately and rigorously? 

Reviewer #1: N/A

Reviewer #3: Yes

4. Have the authors made all data underlying the findings in their manuscript fully available?

Reviewer #1: Yes

Reviewer #3: Yes

5. Is the manuscript presented in an intelligible fashion and written in standard English?

Reviewer #1: No

Reviewer #3: Yes

6. Review Comments to the Author

Reviewer #1: According to the revised paper, I have appreciated the deep revision of the contents and the present form of this manuscript. There is little content, which need be revised according to the comment of reviewer in order to meet the requirements of publish. A number of concerns listed as follows:

（1）In my opinion the current version of the abstract is not suitable. It should clearly state the manuscript main contribution to the body of knowledge, and the motivation that lead the author to this research.

（2） How about the computation complexity of the proposed method.

（3） The contributions should be simplified to highlight your work.

（4）Conclusion: What are the advantages and disadvantages of this study compared to the existing studies in this area?

（5）The inspiration of your work must further be highlighted. Some suggested recent literatures should add according to previous comments.

(6) The paper is in need of revision in terms of eliminating grammatical errors, and improving clarity and readability.

Reviewer #3: The authors made the requested modifications and improved their paper.. I appreciate the detailed responses given to the reviewers. The authors made an important change in the abstract and this is helpful for the readers. Also the references have been corrected to be more suitable to the research. This is an important research and should be continued.

7. PLOS authors have the option to publish the peer review history of their article (what does this mean?). If published, this will include your full peer review and any attached files.

Reviewer #1: No

Reviewer #3: No

---

## [Author Response · Author response to Decision Letter 2]

17 Aug 2022

Reviewer #1:

Comments:

（1）In my opinion the current version of the abstract is not suitable. It should clearly state the manuscript main contribution to the body of knowledge, and the motivation that lead the author to this research.

（2） How about the computation complexity of the proposed method.

（3） The contributions should be simplified to highlight your work.

（4） Conclusion: What are the advantages and disadvantages of this study compared to the existing studies in this area?

（5） The inspiration of your work must further be highlighted. Some suggested recent literatures should add according to previous comments.

 (6) The paper is in need of revision in terms of eliminating grammatical errors, and improving clarity and readability.

An explanation of modification:

1、The reply to comment (1) is as follows：

 The abstract part is described from three aspects, the first is the description of the importance of fault diagnosis of transmission lines and some current problems in this research. Then, the technical route of the method proposed in this paper is summarized. Finally, the advantages of the proposed method and its contribution to fault diagnosis research are obtained from the experimental results. The details are as follows:

Transmission lines are the most prone to faults in the power transmission system, and high-precision fault diagnosis is crucial for quick troubleshooting. In the current intelligent fault diagnosis research methods, there are problems such as difficulty in accurately extracting fault features, low fault identification accuracy and low fault tolerance. In order to solve these problems, this paper proposes an intelligent fault diagnosis method for high-voltage direct current transmission lines (HVDC) based on Gram angle field difference field (GADF) and improved convolutional neural network VGG16. This method first performs variational modal decomposition (VMD) on the original fault voltage signal, and then uses the correlation coefficient method to select the appropriate intrinsic mode function (IMF) component, and converts it into a two-dimensional image using the Gramian Angular Difference Field(GADF). Finally, the improved VGG16 network is used to extract and classify fault features adaptively to realize fault diagnosis. In order to improve the performance of the VGG16 fault diagnosis model, batch normalization, dense connection and global average pooling techniques are introduced. The comparative experimental results show that the model proposed in this paper can deeply mine fault features and has a high fault diagnosis accuracy. In addition, the method is not affected by fault type, transition resistance and fault distance, has good anti-interference ability, strong fault tolerance, and has high potential in practical applications

2、The reply to comment (2) is as follows：

 Judging the pros and cons of an algorithm is mainly measured from two aspects: the execution time of the algorithm (time complexity) and the storage space (space complexity) that it needs to occupy

1)The time complexity of the VMD algorithm can be expressed as: Time~O(n) , where n denotes the number of program executions, and in this paper n=k, k denotes the number of IMF components of the vmd decomposition. 

2)The time complexity of the GADF algorithm can be expressed as Time~O(n) : , where n is the number of times the program is executed, and in this paper n is equal to the number of samples.

3)Time complexity of the convolutional neural network VGG16.

In a convolutional neural network, The computing time is mainly spent on the convolution operation and the operation of the fully connected layer, and the operation of the fully connected layer accounts for most of the time. The greater the number of operations, the greater the computational resources consumed and the longer the time spent. In this paper, in order to reduce the time complexity of VGG16, the global average pooling layer is used instead of the fully connected layer, which greatly reduces the amount of computation in the network, i.e. greatly reduces the time complexity of the network.

The time complexity of a single convolutional layer expressed as (11): 

 Time~O(M²·k²·Cin·Cout) （11） 

 M=(X-K+2*Padding)/ Stride+1 (12)

Where M represents the size of the output feature map of each convolution kernel, and its expression is as in Equation (12)，which is determined by the four parameters of input matrix size X, convolution kernel size K, padding, and stride; In the Equation (11), the k denotes the edge length of each convolutional kernel ; Cin denotes the number of channels of each convolutional kernel, also represents the number of input channels; and Cout denotes the number of convolutional kernels , also represents the number of output channels. It can be seen that the time complexity of each convolutional layer is fully determined by the output feature area M², the convolutional kernel area k², and the number of input and output channels.

 The overall time complexity of the VGG16 network can be expressed as (13):

 Time~O(∑l Ml²·kl²·Cl- 1·Cl+∑j Cj- 1·Mj²·mj ) (13) 

Where D denotes the number of all convolutional layers of the neural network,also represents the network depth; l denotes the first convolutional layer of the neural network; Cl denotes the number of output channels Cout of the lth convolutional layer of the neural network, also represents the number of convolutional kernels in that layer. DFC denotes the number of fully connected layers, mj denotes the number of features to be output in that layer.

 In this paper, the global average pooling layer is used to replace the fully connected layer, so the overall time complexity of the VGG16 network proposed in this paper can be expressed as (14):

 Time~O(∑l Ml²·kl²·Cl- 1·Cl+Cj- 1·mj ) （14）

4)Space Complexity of convolutional neural network VGG16:

 The space complexity is a measure of the size of the storage space temporarily occupied by an algorithm during the running process. which can be expressed as (15) :

 Space~O(∑l K2·Cl- 1·Cl+∑l M2·Cl) （15）

In the ablation experiment, it can be seen that the improved VGG16 algorithm proposed in this paper has a network parameter size of 69MB after removing the fully connected layer. Compared with the unimproved VGG16, it can be seen that the memory occupied by the network parameters is reduced by 179MB. It can be seen that the improved VGG16 has greatly improved the space complexity.

3、The reply to comment (3) is as follows：

The author has refined the contribution of this paper, and summarized it from two aspects of data processing and network improvement, as follows:

(1)Based on the GADF algorithm, the one-dimensional fault signal is converted into a two-dimensional color image, aiming to use deep learning to mine deeper fault feature information.

(2)The batch normalization algorithm is introduced to effectively speed up network convergence and prevent overfitting. The dense connection method of the convolutional layer is designed to speed up the network training and convergence speed while realizing feature reuse and enhancement. Aiming at the slowing down of the calculation speed caused by the large amount of network parameters, a global average pooling algorithm is introduced, which effectively reduces the amount of network parameters.

(3)The fault diagnosis model of GADF-VGG16 is proposed , which improve the fault diagnosis accuracy, and the superiority of the proposed algorithm is verified by an example.

4、The reply to comment (4) is as follows：

The author revised the conclusion part, highlighting the advantages and disadvantages of the proposed algorithm compared with the existing research. The specific revisions are as follows:

Aiming at the problem that the existing fault diagnosis methods are difficult to accurately extract fault characteristics when faced with complex fault characteristics, this paper proposes a fault diagnosis method for HVDC transmission lines based on GADF-VGG16. The algorithm proposed in this paper breaks the traditional thinking, converts the one-dimensional fault signal into a two-dimensional image for self-adaptive extraction of deep fault features, and better combines the advantages of convolutional neural network in image processing to realize intelligent fault diagnosis of HVDC transmission lines. The simulation results show that the proposed algorithm has the following advantages:

（1）The method proposed in this paper uses GADF to convert one-dimensional fault data into two-dimensional color images, retains the temporal correlation in the data, and uses deep learning algorithms to achieve high-dimensional feature extraction. Compared with the method of converting the data into a two-dimensional grayscale image in the literature [18], the data processing method proposed in this paper will not cause the loss of characteristic data, so that the fault diagnosis model has a higher fault identification accuracy.

（2）In this paper, the VGG16 network is improved by adding BN layer and convolutional layer of densely connected structure to speed up the training and convergence speed of the network while realizing feature reuse and enhancement; at the same time, the global average pooling layer is used to replace the fully connected layer, reducing The amount of model parameters and computation time make the proposed method more suitable for rapid fault diagnosis.

（3）Compared with traditional intelligent algorithms, the fault diagnosis model based on improved VGG16 constructed in this paper can fully extract deep fault features. Compared with the fault diagnosis algorithm based on SVM, BP, RF and other shallow neural networks in literature [9-12], the algorithm proposed in this paper has higher fault recognition accuracy, and the experimental results show that it is not affected by transition resistance, fault type and fault. It has strong anti-interference ability and fault tolerance.

On the other hand, the method proposed in this paper also has some limitations. Due to the huge size of the actual HVDC transmission system, the fault situation of the transmission line is more complicated. Limited to laboratory conditions, the fault diagnosis models proposed in this paper are all carried out in a simulation environment, and there is a lack of fault data in the actual operation process to verify the proposed model.

In order to further improve the fault diagnosis model of HVDC transmission lines, how to verify the proposed diagnosis model in combination with the actual fault data of HVDC transmission lines is the focus of the next research work.

5、The reply to comment (5) is as follows：

 In order to highlight the source of inspiration for this study, some recent literatures have been added to the introduction and main text. Based on previous research, the research ideas of this paper are obtained. The specific revisions are as follows:

As a research hotspot in recent years, deep learning has been widely used in image processing, target detection, fault diagnosis and other fields. As Jin et al. [13] introduced an intelligent fault diagnosis method for train axle box bearings based on parameter optimization of VMD and improved DBN. The method uses the gray wolf optimization algorithm (GWO) to optimize VMD and DBN, and solves the parameters of VMD. The problem is set, and VMD is used to decompose the signal, and the feature information of the modal component with the largest correlation coefficient is extracted through the multi-scale distribution entropy. Finally, DBN is used to identify the characteristic information to realize the weak fault diagnosis of bearings.Reference [14] proposed an intelligent diagnosis algorithm based on continuous wavelet transform and Gaussian convolution deep belief network. In this method, the one-dimensional vibration signal is converted into a two-dimensional image and the deep belief network is used for feature extraction, which avoids the artificial extraction of fault features. influences. Smart algorithms have achieved significant improvements over traditional algorithms.Reference [15] proposed a new method for intelligent fault diagnosis based on time-frequency images and deep learning. This method uses short-time Fourier transform to obtain time-frequency images, and combines the multi-sensory data fusion method of deep residual network to carry out the analysis of machine bearings. For fault diagnosis, compared with a single type of fault signal, this method can achieve better diagnostic accuracy.The development of deep learning has also brought new ideas to the field of transmission line fault diagnosis.Reference [16] proposes a fault diagnosis method for HVDC systems based on stacked sparse autoencoders, which utilizes stacked sparse autoencoders for automatic feature extraction, classification and identification. This method makes full use of the self-learning ability of the deep learning algorithm to realize the accurate identification of regional faults.Zhai et al [17] proposed to construct a fault diagnosis model for HVDC transmission lines using an improved convolutional neural network (CNN) to extract features and implement fault classification for current timing data.Compared with the traditional CNN network, the method has a certain improvement in recognition accuracy, but the CNN is not ideal for feature extraction of time-series signals.Therefore, drawing on the idea of converting time series signals into two-dimensional images and combining deep learning algorithms in the field of bearing fault diagnosis[14/15/24], Many scholars have begun to try to use this research method in fault diagnosis of HVDC transmission lines.For example, Wang Jun et al. [18] proposed to convert one-dimensional time series signals into two-dimensional grayscale images, and then use CNN to classify transmission line faults. However, the process of converting the time series signal into grayscale image in this method results in the loss of characteristic data. It can be seen that it is feasible to apply deep learning to fault diagnosis of transmission lines, and further in-depth research is needed.

In view of the advantages and disadvantages of the above fault diagnosis algorithms, this paper adopts the deep learning method to propose a fault diagnosis model for HVDC transmission lines based on GADF-VGG16. The fault voltage signal of the HVDC transmission line is decomposed into modal components by VMD, and the selected IMF modal components are converted into color images through the Gramian Angular Difference Field (GADF), and the images are input into the improved VGG16 for feature extraction and classification.This method uses a novel GADF encoding method for data preprocessing, and constructs a bijective map for one-dimensional time series and two-dimensional space series, which will not cause the loss of feature information[19].At the same time, this paper improves the traditional VGG16 model structure, increases the BN layer and the convolution layer of the dense connection structure, speeds up the training and convergence speed of the network, and realizes the reuse and enhancement of features.At the same time, the global average pooling layer is used to replace the fully connected layer, which reduces the amount of model parameters and computing time, making the proposed method more suitable for rapid fault diagnosis.The experimental results show that the fault diagnosis model proposed in this paper has high accuracy, is not affected by fault type, transition resistance and fault distance, and has good anti-interference ability and fault tolerance.

The added literature:

[13] Zhenzhen Jin, Deqiang He, Zexian Wei,Intelligent fault diagnosis of train axle box bearing based on parameter optimization VMD and improved DBN,Engineering Applications of Artificial Intelligence,Volume 110,2022,104713,ISSN 0952-1976,https://doi.org/10.1016/j.engappai.2022.104713.

[14] H. Zhao et al., "Intelligent Diagnosis Using Continuous Wavelet Transform and Gauss Convolutional Deep Belief Network," in IEEE Transactions on Reliability, 2022, doi: 10.1109/TR.2022.3180273.

[15] Gültekin, Ö., Çinar, E., Özkan, K. et al. A novel deep learning approach for intelligent fault diagnosis applications based on time-frequency images. Neural Comput Applic 34, 4803–4812 (2022). https://doi.org/10.1007/s00521-021-06668-2

[16] X. Wang, D. Zhang, X. Zhang and P. Ni, "Fault diagnosis method for hybrid HVDC transmission system based on stacked sparse auto-encoder," 2022 7th Asia Conference on Power and Electrical Engineering (ACPEE), 2022, pp. 1771-1776, doi: 10.1109/ACPEE53904.2022.9783919.

[24] TONG yu, PANG xinyu, WEI zihan. Fault diagnosis method of rolling bearing based on GADF-CNN[J]. Vibration and shock, 2021,40(05):247-253+260.

[26] Yang, H., Ni, J., Gao, J. et al. A novel method for peanut variety identification and classification by Improved VGG16. Sci Rep 11, 15756 (2021). https://doi.org/10.1038/s41598-021-95240-y

Explanation of the added literature:

 Reference [13] used VMD to decompose the signal, and extracted the feature information of the modal component with the largest correlation coefficient through the multi-scale distribution entropy. This paper uses the data processing method of [13] to preprocess the fault signal. First, the fault signal is decomposed by VMD to obtain K modal components, and then the correlation coefficient method is used to select the modal component that is most similar to the original signal as the characteristic signal. In the field of bearing fault diagnosis, literature [14], [15], [24] and others proposed a data processing method to convert one-dimensional data into two-dimensional images. The GADF coding method is novel, and the bijective mapping is constructed in one-dimensional time series and two-dimensional space series, which will not cause the loss of feature information. Therefore, this paper adopts GADF encoding method to convert the preprocessed IMF components into two-dimensional color images.The introduction of deep learning has brought new ideas to the field of transmission line fault diagnosis. For example, [16] made full use of the self-learning ability of deep learning and proposed to use DBN to extract and identify fault features. In view of the remarkable achievements of CNN in the field of image classification, this paper draws on the excellent VGG16 model in the literature [26] as the feature extraction model of this paper, and makes a series of improvements to the original VGG16 model to make it more suitable for HVDC transmission lines. fault diagnosis.

6、The reply to comment (6) is as follows：

The author has carefully checked the text for grammatical errors and corrected them to improve the clarity and readability of the text.

Reviewer #3:

Comment :

The authors made the requested modifications and improved their paper.I appreciate the detailed responses given to the reviewers. The authors made an important change in the abstract and this is helpful for the readers. Also the references have been corrected to be more suitable to the research. This is an important research and should be continued.

An explanation of modification:

 I am very grateful to the reviewers for their recognition. The authors have made further revisions for the shortcomings of the article. I hope that the reviewers will criticize and correct the revised manuscript submitted by the authors.

---

## [Decision Letter · Decision Letter 3]

1 Sep 2022

GADF-VGG16 based Fault diagnosis method for HVDC transmission lines

PONE-D-22-08053R3

Dear Dr. wu,

We’re pleased to inform you that your manuscript has been judged scientifically suitable for publication and will be formally accepted for publication once it meets all outstanding technical requirements.

Kind regards,

Ashwani Kumar, Ph.D.

Academic Editor

PLOS ONE

Additional Editor Comments (optional):

Reviewers' comments:

Reviewer's Responses to Questions

**Comments to the Author**

1. If the authors have adequately addressed your comments raised in a previous round of review and you feel that this manuscript is now acceptable for publication, you may indicate that here to bypass the “Comments to the Author” section, enter your conflict of interest statement in the “Confidential to Editor” section, and submit your "Accept" recommendation.

Reviewer #1: (No Response)

Reviewer #4: (No Response)

2. Is the manuscript technically sound, and do the data support the conclusions?

Reviewer #1: (No Response)

Reviewer #4: Yes

3. Has the statistical analysis been performed appropriately and rigorously? 

Reviewer #1: (No Response)

Reviewer #4: Yes

4. Have the authors made all data underlying the findings in their manuscript fully available?

Reviewer #1: (No Response)

Reviewer #4: Yes

5. Is the manuscript presented in an intelligible fashion and written in standard English?

Reviewer #1: (No Response)

Reviewer #4: Yes

6. Review Comments to the Author

Reviewer #1: I have appreciated the deep revision of the contents and the present form of this manuscript. All my previous concerns have been accurately addressed. I think that this paper can be accepted.

Reviewer #4: It is better to have plots for convergence and comparison plots, which will make the work more validated.

7. PLOS authors have the option to publish the peer review history of their article (what does this mean?). If published, this will include your full peer review and any attached files.

Reviewer #1: No

Reviewer #4: **Yes: **S.N.Deepa

---

## [Editor Report · Acceptance letter]

8 Sep 2022

PONE-D-22-08053R3 

GADF-VGG16 based fault diagnosis method for HVDC transmission lines 

Dear Dr. Wu:

I'm pleased to inform you that your manuscript has been deemed suitable for publication in PLOS ONE. Congratulations! Your manuscript is now with our production department. 

Kind regards, 

on behalf of

Dr. Ashwani Kumar 

Academic Editor

PLOS ONE